# System Dynamics Analysis of the Relationship between Transit Metropolis Construction and Sustainable Development of Urban Transportation—Case Study of Nanchang City, China

**Yunqiang Xue** [1,2,3,*] **, Lin Cheng** [2] **, Kuang Wang** [1] **, Jing An** [1] **and Hongzhi Guan** [1,4]

[1]   College of Transportation and Logistics, East China JiaoTong University, Nanchang 330013, China;
      wangkuang0320@163.com (K.W.); anjing@ecjtu.edu.cn (J.A.); hguan@bjut.edu.cn (H.G.)
[2]   School of Transportation, Southeast University, Nanjing 210096, China; gist@seu.edu.cn
[3]   High-Speed Rail and Regional Development Research Center of Jiangxi Province, Nanchang 330013, China
[4]   College of Architecture and Civil Engineering, Beijing University of Technology, Beijing 100124, China
*   Correspondence: xueyunqiang@ecjtu.edu.cn; Tel.: +86-132-417-324-69

**Abstract:** In order to systematically analyze the benefits of transit metropolis construction, the system dynamics (SD) theory was used to construct the transit metropolis SD simulation model from the four subsystems of economy, society, environment, and transportation supply and demand. The validity of the SD model was verified by the social and economic data of Nanchang City and the operational data of the bus company, and the quantitative simulation analysis was carried out by taking the construction of the transit metropolis in Nanchang as an example. The simulation results show that, in 2020, the number of motor vehicles in Nanchang will reach 1.13 million and the urban population will reach 5.71 million. It is necessary to build a transit metropolis for the sustainable development of urban transportation. In order to complete the transit metropolis creation goal of 60% of the public transit mobility sharing rate, the proportion of public transport investment in the total transportation investment needs to be adjusted from 0.25 to 0.35. As a result, Nanchang City will improve after the peak traffic congestion in 2022, indicating that the construction of the transit metropolis will have a positive effect on Nanchang. By developing new energy vehicles and low-emission vehicles, vehicle emissions will drop from 0.05 tons/year to 0.04 tons/year, and overall nitrogen oxide emissions will fall by 70%, which is significant for urban environments. The research results provide theoretical support for the significance of transit metropolis construction, and promote the sustainable development of urban transportation.

**Keywords:** transit metropolis; system dynamics; sustainable development; urban transportation; modelling and simulation

## 1. Introduction

With the rapid development of China's economy, society, and urbanization, the number of motor vehicles has increased dramatically, bringing urban traffic problems such as traffic congestion, traffic accidents and environmental pollution [1]. According to the data released by the Traffic Management Bureau of the Ministry of Public Security of China, at the end of 2019 the number of motor vehicles in the country reached 348 million, including 260 million cars and 207 million private cars, with 435 million motor vehicle drivers, of which 397 million were motorists. The number of owners and drivers will continue to increase. The National New Urbanization Plan (2014–2020) proposes that China's

urbanization rate will further increase to 60% in 2020. With the further increase of the urban population and the number of motor vehicles, urban transportation problems will become more and more serious.

Compared with cars, public transportation has obvious advantages due to its per capita occupation of road resources and per capita energy consumption [2]. Public transportation has become an effective travel tool for urbanization and the sustainable development of urban transportation [2–4], and it is common sense to address urban transportation issues by prioritizing the development of public transportation [5,6]. The concept of public transport priority originated in France in the 1960s and was subsequently improved in developed countries such as Great Britain, the United States, and Japan [7–9]. China has advocated public transport priority since the 1980s. Determining public transport priority as a national level development strategy is the subject of Circular no. 64, issued by the State Council in 2012, titled "Guiding Opinions on Urban Priority Development of Public Transport." In order to cope with urban traffic problems and implement Circular no. 64, the Ministry of Transport (MoT) launched the Transit Metropolis Program in 2012.

Up to now, a total of three batches of 87 cities have been selected as Transit Metropolis creation demonstration projects. Fourteen cities, including Shanghai and Nanjing, have completed the establishment of transit metropolises and officially become National Transit Metropolis Demonstration Cities. The city applying for the Transit Metropolis Program needs to meet the requirements of 20 assessment indicators (Table 1) and 10 reference indicators (Table 2) [10], and propose specific quantitative and qualitative goals for its own city. A city formally wins the honorary title of National Transit Metropolis Demonstration City only after at least five years of the creation and successful completion of the creation goals proposed in the application. In addition, the city applying for the Transit Metropolis Program can also propose no more than three characteristic indicators based on its own characteristics. For example, due to its abundant underground spring water, Jinan City lacks large-capacity subway lines; the BRT (Bus Rapid Transit) network forms the backbone of the public transportation network. Therefore, Jinan takes the ratio of the length of the BRT line network to the length of the entire public transportation line network as a characteristic indicator of the creation of a transit metropolis. In fact, Curitiba, Brazil, is the hometown of BRT. Many cities in the world, including most large and medium-sized cities in China, have planned and built BRT under the guidance of the Brazilian experts from the World Bank. The BRT system also plays an important role in the construction of a transit metropolis. Robert Cervero [11,12] detailed the efficient BRT system in Curitiba, Brazil, in his book *Transit Metropolis*.

**Table 1.** Assessment indicators for the creation of a transit metropolis.

| Serial Number | Assessment Indicator | Serial Number | Assessment Indicator |
|---|---|---|---|
| 1 | Motorized travel sharing rate of public transport | 11 | Green bus vehicle ratio |
| 2 | Ratio of bus line network | 12 | Fatality rate of bus-involved traffic accidents |
| 3 | 500-m coverage at bus stations | 13 | Fatality rate of rail-involved traffic accidents |
| 4 | Public transport vehicle ownership per 10,000 residents | 14 | Ratio of transit operation in urban–rural passenger transport |
| 5 | Punctuality of public transport | 15 | Public transport operation subsidy system and subsidy availability rate |
| 6 | Average speed of public buses and trams in morning and evening peak hours | 16 | Usage rate of public transport smart card |
| 7 | Average crowded level of public transport in morning and evening peak hours | 17 | Inter-provincial public transport card usage |
| 8 | Public transport passenger satisfaction | 18 | Construction and operation of intelligent public transport system |
| 9 | Rate of entering parking lot of buses and trams | 19 | Urban public transport planning and implementation |
| 10 | Bus lane setting rate | 20 | Implementation of traffic impact assessment for construction projects |

**Table 2.** Reference indicators for the creation of a transit metropolis.

| Serial Number | Assessment Indicator | Serial Number | Assessment Indicator |
|---|---|---|---|
| 1 | Public transport sharing rate without walking | 6 | Average station area of buses and trams |
| 2 | Public transport trips per capita per day | 7 | Bus bay setting rate |
| 3 | Density of bus network | 8 | Ratio of bus signal priority intersections |
| 4 | Average age of buses and trams | 9 | Income level of public transport employees |
| 5 | Settlement rate of public transport complaint events | 10 | Formulation of supporting policies for priority development of public transportation |

As an urban transportation strategy adopted in response to the rapid growth of cars and traffic congestion, the transit metropolis will promote the priority of public transportation and sustainable development of urban transportation in China to a new level. However, if the establishment of a transit metropolis is completed, how much will it promote the sustainable development of urban transportation? How to systematically evaluate the role of a transit metropolis in promoting sustainable development of urban transportation? All the questions are worth exploring.

## 2. Literature Review

### 2.1. Research Status of the Transit Metropolis

The term transit metropolis was first proposed by Robert Cervero of the University of California, Berkeley, a well-known transportation expert, in his book *Transit Metropolis* [11]. Transit metropolis refers to an area where public transportation services and urban forms develop harmoniously. It advocates urban public transportation to actively guide urban development and emphasizes the coexistence of urban public transportation and the city [12]. The transit metropolis generally has a high share of urban public transport, compact urban space layout, a diversified urban public transport service network, a people-oriented public transport priority policy, and efficient integrated urban transport management. Tokyo, Seoul, Singapore, and Hong Kong in Asia, and London, Paris, Stockholm, and Copenhagen in Europe are the world's eight major transit metropolises [11]. The book *Transit Metropolis* provides a wealth of insights about transportation problems and practical solutions and is becoming an important resource for planners and managers throughout the world [13]. The Transit Metropolis Program proposed by the Ministry of Transport of China has established a set of public transport city assessment index systems in line with national conditions [13]. Carrying out the demonstration project of Transit Metropolis construction in China is a major measure to implement the strategy of prioritizing public transportation development, regulate and guide traffic demand, alleviate urban traffic congestion and pressure on resources and environment, and promote the sound and rapid development of urban public transportation in the new era [10]. The impact of the Transit Metropolis Program will be profound [10–15]. First, it is an important vehicle for implementing the strategy of prioritizing the development of urban public transportation. The central task of the construction of a transit metropolis is to fully mobilize enthusiasm in all aspects, provide power and create experience for the comprehensive implementation of the priority development strategy of public transportation, comprehensively improve the service quality and security capabilities of public transportation, and fundamentally change the lagging development and passive adaptation of urban public transportation. Secondly, it includes specific actions to protect and improve people's livelihoods. The important goal of the construction of a transit metropolis is to protect the basic travel rights of the people. This is an important task for the transportation sector in order to strengthen and innovate social management. Thirdly, it is an important way to change the development mode of urban traffic and an effective way to manage urban traffic congestion. The essence of Transit Metropolis construction is strategically guided by "public transportation-oriented urban development." Through scientific planning and system

construction, a public transportation-based urban transportation system is established to reverse the passive adaptation of urban public transportation to urban development and realize positive interaction and coordinated development of public transportation and the city. Fourthly, it is an effective way to alleviate urban traffic congestion. Urban traffic congestion has become a prominent problem commonly faced by large and medium-sized cities in China and a hot spot that widely concerns all sectors of society. The experience of countries around the world shows that focusing on urban scientific planning and prioritizing the development of public transportation are the most fundamental and effective ways to alleviate urban traffic congestion. The core of the construction of a transit metropolis is to continuously improve the attractiveness of the city's public transportation system and reduce the residents' dependence on cars through the implementation of scientific planning and control, the optimization of networks, facilities construction, information services, and comprehensive management. It would regulate the total amount of urban traffic demand and travel structure, improve the efficiency of urban traffic operation, and fundamentally alleviate urban traffic congestion.

Jiangping Zhou [10,14] reviewed and surveyed characteristics of the Chinese transit metropolis and compared the contents of the transit metropolis proposed by Cervero and by the MoT, respectively. The performance measures for a transit metropolis proposed by the Ministry of Transport of China are more universal and quantitative than Cervero's, the policies and perspectives are still parochial [14]. More work is needed to better define, analyze, and implement a transit metropolis in China. Tian Jiang [15] assessed the status of public transport in Chinese major cities and analyzed the factors that affect the public transport development supporting the Transit Metropolis Program. With the growth of GDP, the public transit share rate tends to go "up-down-up" at different stages of the economy. Chen and Yang [14,16] argued that the definition of transit metropolis given by Cervero is appropriate and qualitative; they added goals and objectives to expand Cervero's definition. Liu Lan [17] proposed that quantitative indicators such as level of congestion, average vehicle speed, on-schedule rate, ratio of urban and rural public transit services should be used to evaluate transit metropolises in different aspects. His indicators are similar to those of the Ministry of Transport of China [14]. The MoT's Transit Metropolis focuses on government support, accessibility, safety, availability, level of services, technologies, and internal employee retention of public transport [14], while Cervero focuses on how to realize the strong connection between public transit and land use [12]. The former has more detailed, stringent, and universal requirements; for instance, the motorized travel sharing rate of public transport should not be less than 60%, and 500-m coverage at bus stops and subway stations should comprise at least 90% of built-up areas. The MoT cares more about service levels, availability, accessibility and scope of coverage of public transport; it ignores land use patterns, housing, and activities around the bus stop and subway station or along transit corridors. Although there are differences in transit metropolis and measurement indicators at home and abroad, there is a consensus that a transit metropolis should embody a structure of urban layout that uses urban public transportation as the main body of motorized travel and guides urban development with urban public transportation. Constrained by safety, resources, environment, and other conditions, the best form of urban construction is an urban development model with the best overall efficiency and social and environmental benefits. The transit metropolis advocates urban public transportation to actively guide urban development, emphasizing that urban public transportation and urban human settlements, structural functions, the environment, and spatial layout should coexist harmoniously. The transit metropolis is a leap in understanding the theory and practice to solve urban and transportation problems.

The construction of the transit metropolis has been a great opportunity for public transport development in recent years. The application cities will further expand the scale based on the current 87 cities. With the rapid development of transit metropolises, fruitful research results have been published and are a hot topic. Yu Qiaolan [18] compared the differences between the two concepts of bus priority and transit metropolis from the three aspects of connotation, implementation, and strategy, and used the specific city of Shanghai to further elaborate the transit metropolis concept. Ding Chuan [19] discussed the relationship between transit metropolitan strategy and the TOD (Transit-oriented Development)

model from the perspective of low-carbon travel. An Meng [20] analyzed the differences between transit metropolis and bus priorities from four aspects: meaning, measures, indicators and goals, and provided guidance for cities to create transit metropolises. Yu Hao [21] took Nanjing as an example to discuss the bus-oriented land-use model in the process of transit metropolis construction, and analyzed the relationship between bus development and land use.

Nowadays, China is the only country in the world that adopts a top-down program to actively promote transit metropolis nationwide [14]. The related researches and ongoing transit metropolis efforts in China will help to generate more benefits for sustainable transportation and promote more transit metropolises in the future. China's Transit Metropolis Program construction experience will provide reference for other cities in the world to develop public transit and promote urban sustainable transportation development. The 14 National Transit Metropolis Demonstration Cities, including Shanghai and Nanjing, have established the dominant position of public transportation in the urban transportation system, and the role of public transportation in urban development has been significantly enhanced, which better meets the basic travel needs of the general public. In these National Transit Metropolis Demonstration Cities, the share of public transport motorized trips is above 60%, and the density of public transit networks in urban built-up areas is more than 3 km/km$^2$; the coverage rate of public bus stops and subway stations in urban built-up areas reaches more than 90% of passengers, who can get on the bus within 500 m and transfer in 5 min in the main urban area; the satisfaction rate of passengers of urban public transport is higher than 80%; the average annual mortality rate of public traffic accidents on public cars and trams is controlled to within 4.5 people per 10,000 standard vehicles; the planning system for urban public transportation has been formed The system of urban public transport policies and standards is basically complete; the integrated management pattern of urban and rural passenger transport has been basically formed. All these demonstration cities have built a well-defined public transportation network consisting of express lines, trunk lines, branch lines, micro circulation lines, and customized bus lines; metro networks, bus networks, and slow transit networks have been formed.

### 2.2. Research on System Dynamics in Traffic Problems such as Transit Metropolis Construction

The construction of a transit metropolis is a systematic project, and the public transport system itself is also a complex system with multiple feedbacks, variables, and nonlinearities [22]. The primary motivation for adopting system dynamics (SD) as the methodology is that the research needs to represent concurrent and multiple intersections among variables in different subsystems. The SD approach can allow one to understand and interpret the interactions easily [22]. Furthermore, a key strength of the SD approach is that it can describe the dynamic processes which evolve continuously and with lags or time delays. This ability is important since we need to study the cumulative impacts of transit metropolis construction over years. Finally, the SD model can describe nonlinear relationships among variables [22]. The nonlinear interactions of multiple factors that are parts of decision-making and the basic physics of systems will be potentially ignored if the nonlinear relationships are not considered. The SD approach can suitably describe the cumulative impacts and developing trends of transit metropolis construction. What is more, SD can reflect the variable structure of the complex system of transit metropolis construction and can predict the trend and cumulative effect of system variables over time. It is obvious that to research the transit metropolis program as a whole requires a systematic analysis approach.

In the process of traffic policy formulation, due to the complexity, dynamics, and uncertainty of the transportation system, accurate prediction and analysis are difficult to achieve and unnecessary, and trend analysis is needed more [23]. Compared with traditional qualitative and quantitative analysis methods, the SD method qualitatively analyzes the interaction relationships among system elements and also quantifies these relationships through the construction of dynamic differential equations, and it uses computer simulation technology to achieve the dynamics of the relationships among transportation elements. Simulation is helpful to reflect the effect of policy implementation. It is

obvious the SD method is one of the best methods to analyze the transportation system and evaluate the transportation policy [23]. Therefore, it is applicable and feasible to use the SD method to study traffic problems such as the construction of a transit metropolis, which can reflect the advantages of systemicity and dynamics.

At present, SD is commonly used to systematically study traffic problems, including transportation systems, highways, road traffic emission policies, congestion policies, freight transportation, and so on. Many researchers have used the SD approach to analyze transportation system problems. Wilsses et al. [24] considered it suitable to capture the causality of urban transportation system variables using system dynamics and analyzed the impact of some Brazilian policies on the transportation system environment, economy, and transportation variables. They conducted extreme condition tests and integration error tests for model validation via the software Vinsim. Yongtao Tan et al. [25] used Beijing, China, as an example to simulate the sustainable performance of the city with a system dynamics model. In order to achieve sustainable development, Beijing should adopt a low speed urbanization policy. They performed model verification with historical data such as total population, service industry output value, number of health workers, annual total water consumption, and farmland area. The absolute error rates were less than 5%. The SD approach is also used to research highway problems. Gokhan et al. [26] used a system dynamics method to study highway sustainability and simulated three potential strategies for policy development: fuel efficiency, public transportation, and electric vehicle usage. The mixed implementation of strategies played an important role in the success of policy making. In the process of model validation, the $CO_2$ emission values obtained from the SD model were compared with the real data, and a one-way ANOVA statistical hypothesis test was also used to compare the simulated trend and the actual emission trend [26]. Additionally, the SD method is used to study road traffic emission policies. Aiga Barisa et al. [27] used a system dynamics model to analyze $CO_2$ emission reduction policies of road traffic. The SD model can better understand the factors behind road emissions. Structural and behavioral validation tests were performed for model validation in the research, which compared the simulated values with the real values and examined the sensitivity of the SD model's behavior under the changed values of some major parameters. Xue Liu [28] proposed an SD approach to scenario analysis for energy consumption and $CO_2$ emissions of urban passenger transport and built a Beijing urban passenger transport carbon model. The SD model simulated different policy scenarios under different conditions. What is more, some authors have used SD to analyze congestion policies. Shiyong Liu [29] used the SD model to assess the dynamic impact of congestion pricing strategies on the socioeconomic system of transportation and to support policy makers; the population, employment growth, and tourism are exogenous factors that affect travel demand. For the freight transportation problems, Carina Thaller et al. [30] used the SD model to describe and reveal urban freight traffic, and made medium and long-term predictions. The proposed SD model enabled trend analysis by medium and long-term forecasts [30]. Wen Huang [31] proposed a methodological framework for truck weight regulation based on system dynamics. The framework was composed of five subsystems and captured the vehicle, highway, and freight variables that influence the effects of truck weight regulation and transportation efficiency over time. For model validation, the freight volume, truck traffic, and freight turnover outputs simulated by the SD model were compared to the actual values over the same period. Significant correlations were found between the actual data and the simulated data. The SD model was verified by finding a significant correlation between actual and simulated data [31]. For some other problems, Eirini Grammatiki Pagoni [32] developed a novel SD-based decision support tool to assess the socially and financially sustainable performance of national public–private partnership programs. Historical data were used to estimate parameters of the SD model and their trends, such as Population_Net_Fractional_Rate, Income_Per_Capita, and Labour_Force_Participation. The test results indicated no systematic bias between the simulated and actual values; hence, the SD model could replicate the observed behavior in the studied system [32]. These researches are valuable for understanding the SD model structure, model verification, model simulation and applications in

policy analysis. The Transit Metropolis Program has Chinese characteristics. Although there are many researches on the transit metropolis in China [10,14,15,17–21], there is little research on the systematic effects of the Transit Metropolis Program.

The construction of a transit metropolis is a complex system, including a series of aspects such as environment, urban economy, urban population, and other aspects of transportation demand and supply. This also requires that the construction of a transit metropolis must be considered from multiple aspects. System dynamics is an effective tool for studying complex systems. However, there is little literature on systematically studying the construction of a transit metropolis, especially the quantitative relationship between transit metropolis construction and sustainable development of urban transportation. We hope this systematic quantitative analysis can describe the construction effect of the transit metropolis and illustrate how to take measures to achieve the transit metropolis construction indicators, so as to better promote the sustainable development of urban transportation and meet the citizens' travel needs. In the context of transit metropolis construction, taking Nanchang city as an example, the system dynamics theory is used to quantitatively analyze the progress made in transit metropolis construction, and corresponding suggestions are proposed for the construction of a transit metropolis and the development of public transportation in Nanchang. Due to the similarity of public transit city construction indicators, the analysis ideas and system dynamic models constructed in this paper can be applied to other transit metropolis construction cities.

The research structure is arranged as follows: Section 3 is an introduction to the research methods; SD model construction is presented in Section 4; Section 5 is the SD model simulation, and the last section is the Conclusion and prospects.

## 3. Methodology

This paper adopts a system dynamics method to characterize the construction effect of the transit metropolis. Since Forrester first proposed the system dynamics model in 1961 [33], it has been widely used in various fields of studies.

SD consists of stock, flow, auxiliary variables, and feedbacks [24]. Stock is the variable state in a system that influences system behavior at a certain time. In a period of time, flow is the decrease or increase in the value of a stock. The rates that change stocks during a period of time via determining the flow value are auxiliary variables. The feedback loops are commonly presented in the form of causal relationship diagrams. After the flow, flock, auxiliary variables, and causal relationships are determined and verified, the researched system is simulated with respect to the past data as initial baseline values.

System dynamics modeling includes five steps (Figure 1) [34]. The first step is to clarify the research problems and system boundaries under the guidance of the research goals. The second step is to analyze the system structure, determine the system structure and sub-modules, and sort out the cause and effect of system feedback mechanisms and variables The third step is to establish a dynamic model, including the system flow diagram and dynamic equations. The fourth step is to verify the model structure and parameters until the accuracy requirements are met. Finally, the SD model is used for policy evaluation.

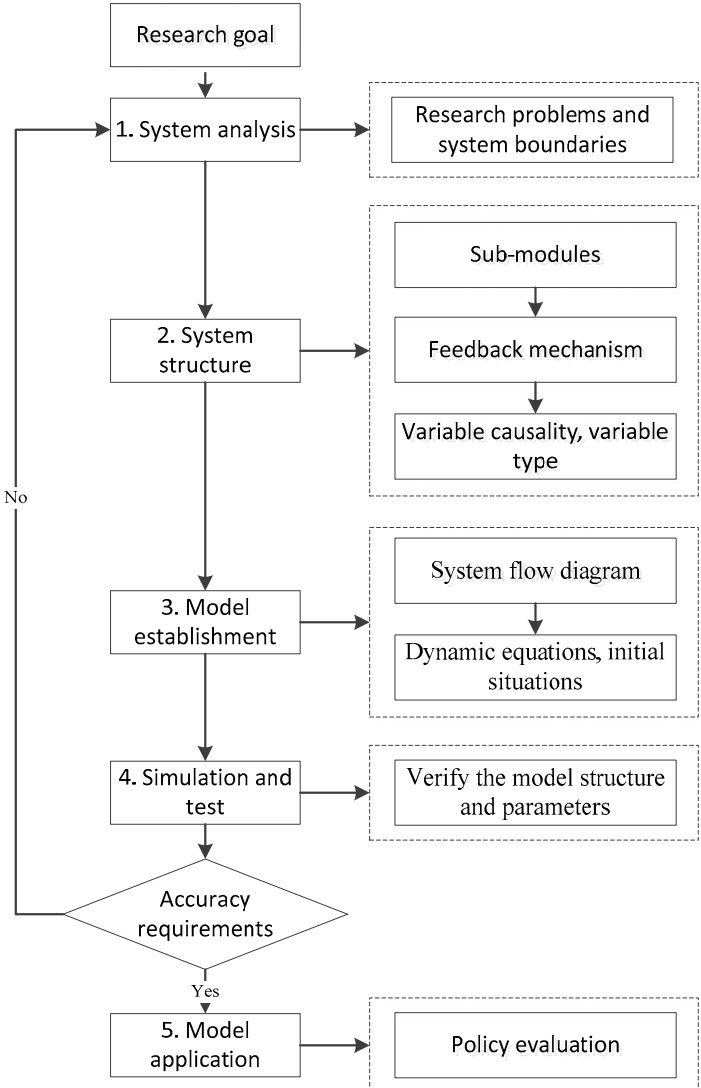

**Figure 1.** Modeling process of the system dynamics (SD) method.

## 4. SD Model of Transit Metropolis Construction

### 4.1. Research Scope and Research Goal

The transportation system includes environmental factors, population factors, economic factors, and transportation supply and demand factors. The transportation supply and demand, social, environmental, economic, and population components of the transit metropolis are selected as the research scope (Table 3). The research objective is to build an SD model of the construction of a transit metropolis, evaluate the effects of the transit metropolis construction strategy, and provide policy recommendations for the realization of the task of a transit metropolis and the promotion of urban sustainable transportation.

**Table 3.** System elements of transit metropolis construction.

| Factors | Specific Parameters | Unit |
| --- | --- | --- |
| Environmental factor | Cumulative NOx emissions | ton |
| | Increase amount of NOx emissions | ton |
| | NOx dissipation rate | —— |
| | NOx dissipation amount | ton |
| | NOx environmental capacity | ton |
| | Average annual NOx emissions per vehicle | ton/vehicle/year |
| | Contribution of motor vehicles to NOx | —— |
| Population factor | Natural growth rate of population | —— |
| | Population migration rate | —— |
| | Growth amount of population | million |
| | Trips per capita | trip |
| | Urban resident population | million |
| | Total resident travel | trip |
| Economic factor | Local GDP | Yuan (RMB) |
| | GDP growth | Yuan (RMB) |
| | GDP growth rate | —— |
| | Investment in transit metropolis | Yuan (RMB) |
| | Proportion of transportation investment in regional GDP | % |
| | Transit investment ratio in transportation investment | % |
| | Transit investment conversion rate | —— |
| | Road investment ratio in transportation investment | % |
| | Road construction investment | Yuan (RMB) |
| | Road investment conversion rate | —— |
| Transportation supply and demand factor | Car trips | trip |
| | Passenger capacity per bus per day | trip/bus/day |
| | Average passenger factor of car | person/car |
| | Bus ownership | standard vehicle |
| | Regular bus trips | trip |
| | Rail transit trips | trip |
| | Road mileage | km |
| | Road capacity | vehicle/hour |
| | Road saturation | —— |
| | Motorized sharing rate of public transport | % |
| | Sharing rate of public transport | % |

## 4.2. System Structure of Transit Metropolis Construction

After determining the scope and boundary of the model, the interactions between the four main boundaries of transportation supply and demand, environment, economy, and population are analyzed. We established a causality diagram for the construction of a transit metropolis, as shown in Figure 2. The causality diagram is also called a system cycle diagram, or causal loop diagram. Starting from the simple causality relationship between system elements, a relationship chain and a causality circuit are established to form a diagram of mutual constraints between the elements [22]. The interrelationships among a series of elements constitute a causal chain or a causal circuit. The causal loops are determined based on the cause and effect relationships among parameters [26]. The plus and minus signs next to the arrows indicate the positive and negative attributes of the correlation between the variables. It should be noted that each chain in the causal circuit diagram must represent a causal relationship between variables, not just a correlation between variables [35,36]. Correlation between variables reflects the past behavior of the system, and correlation does not represent the results of the system. Confounding relationships with causality can lead to misleading and policy errors. Although statistics show a positive correlation between ice cream sales and murder, both of which fluctuate with the average temperature and increase in summer and decrease in winter, the cause and effect chain cannot be drawn between the two. Murder crimes cannot be reduced by limiting consumption of ice cream [36]. The modeler must carefully consider whether the relationship in the dynamic model is causal, regardless of how strong the correlation is, or how statistically important the regression

coefficient is [35,36]. This requires a large collection of historical data, follow-up inspection, statistical inference, and logical experience judgment [36].

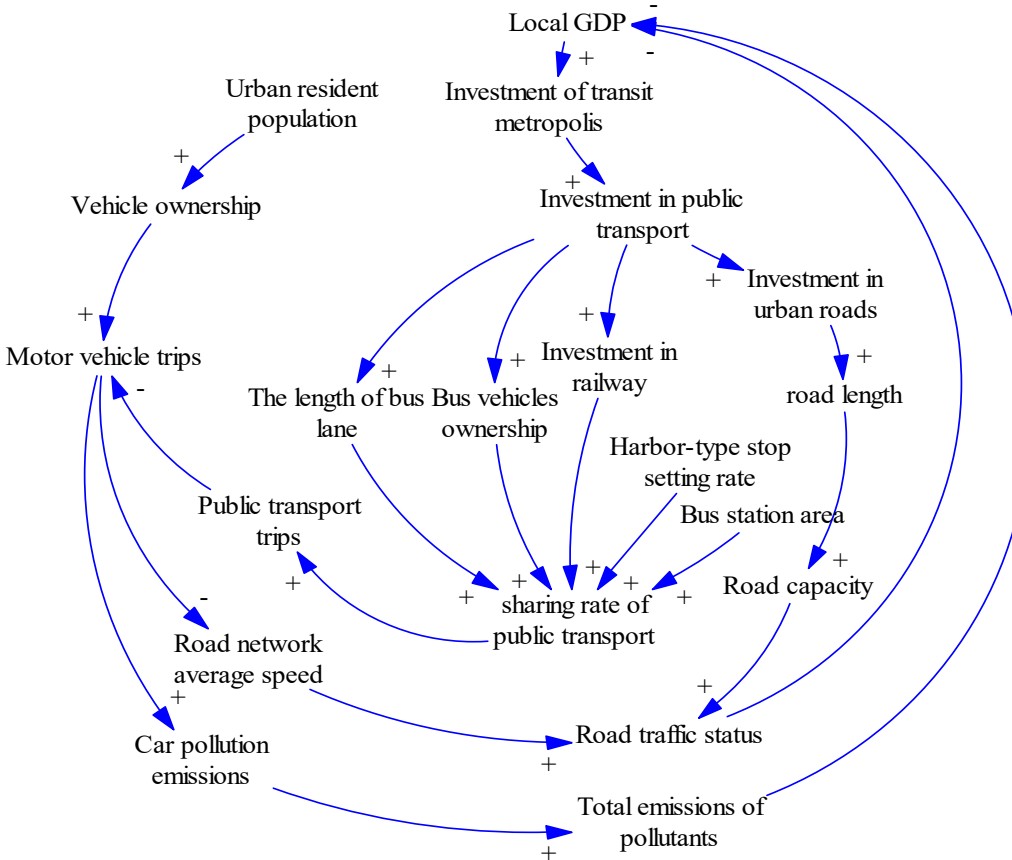

**Figure 2.** Causality diagram of transit metropolis construction.

The main causal loops in the model causality diagram are based on the following logic:

(1) The improvement of regional GDP has a positive effect on the investment of transit metropolis construction. The investment of transit metropolis construction has a positive correlation with the share of public transport, and has a positive correlation with the amount of public bus trips. The increase in the average speed of the road network is conducive to the improvement of road traffic conditions, and the degree of road congestion has an adverse effect on the local GDP.

(2) The growth of the urban economy will lead to an increase in the city's resident population and per capita GDP, and the increase in people's living standards will also increase the number of motor vehicles. This will adversely affect the state of road traffic and the transportation environment, and will adversely affect economic development.

(3) Urban economic development has resulted in a corresponding increase in urban transport investment. Optimization of urban road mileage and road structure has improved road capacity and has a positive effect on urban economic development.

(4) The increase of urban resident population will increase the number of urban residents who travel, which will have an adverse effect on road load and the average speed of the road network. The deterioration of the urban road network has a negative effect on the number of urban residents.

To reiterate, this article studies the relationship between the construction of the transit metropolis and the sustainable development of urban transportation. The transit metropolis construction first requires economic development and capital investment. The sustainability of urban transportation is mainly reflected by road traffic conditions and traffic pollutant emissions. Therefore, the causal loops that include factors such as GDP, urban investment, road traffic conditions, and traffic emissions are

the most critical. Starting with GDP, three causal loops are considered, and one of them is reinforcing and the other two are balancing.

Loop 1: Local GDP → +Investment of transit metropolis → +Investment in public transport → +Investment in urban roads → +road length → +road capacity → +road traffic status → -Local GDP (Balancing loop)

Loop 2: Local GDP → +Investment of transit metropolis → +Investment in public transport → +The length of bus lane/Bus vehicles ownership/Investment in railway → +Sharing rate of public transport → +Public transport trips → -Motor vehicle trips → -Road network average speed → +Road traffic status → -Local GDP (Balancing loop)

Loop3: Local GDP → +Investment of transit metropolis → +Investment in public transport → +The length of bus lane/Bus vehicles ownership/Investment in railway → +Sharing rate of public transport → +Public transport trips → -Motor vehicle trips → +Car pollution emissions → +Total emissions of pollutants → -Local GDP (Reinforcing loop)

These causal relationships seem to hold true, and a model test is needed to compare model simulation values with real values. In fact, the SD model has a delay phenomenon. Delay is a process whose output lags behind its input in some modes. For example, the investment effect of the transit metropolis will have a time period. Investment and construction of bus lanes and subways require a series of times for bidding, construction, supervision, and operation; the purchase of bus vehicles also requires a time for purchase and vehicle filing. For simplicity, this article ignores these delays, and they need to be considered further in future research.

Causality diagrams describe the interrelationships between variables, and no quantitative analysis has yet been given. The quantitative relationships between the variables will be analyzed in the next section, when discussing how a change in a variable affects the change in the system.

### 4.3. System Flow Diagram and Dynamics Equations of Transit Metropolis Construction

According to the cause–effect diagram, the system flow diagram (Figure 3) for the construction of a transit metropolis is divided into four subsystems: economy, society, environment, and transportation supply and demand. The system flow diagrams and dynamic equations of these four subsystems are given below. The symbol  means a rate variable, a cloud  represent a source, the starting point of the rate variable. The black arrow points to a state variable. For example, population growth is a rate variable, the source of population is the initial value and it will change with the time, at the same time it points to a variable urban resident population. Some table functions are based on the *Nanchang Economic and Social Statistical Yearbook*. Therefore, if the SD model is used to describe transit metropolis construction for other cities, the data of the table functions need to refer to the historical data of the local city. Some variables are general for other cities.

### 4.3.1. Flow Chart and Dynamic Equation of the Economic Subsystem

The economic system includes a state variable regional GDP (Local GDP), the rate variables include: GDP growth, raw GDP growth, GDP growth rate, public transport investment conversion rate, urban road investment conversion rate, and the impact factor of traffic congestion as auxiliary variables (Figure 4). There are two table functions in the economic subsystem, the original GDP growth and the traffic congestion impact factor.

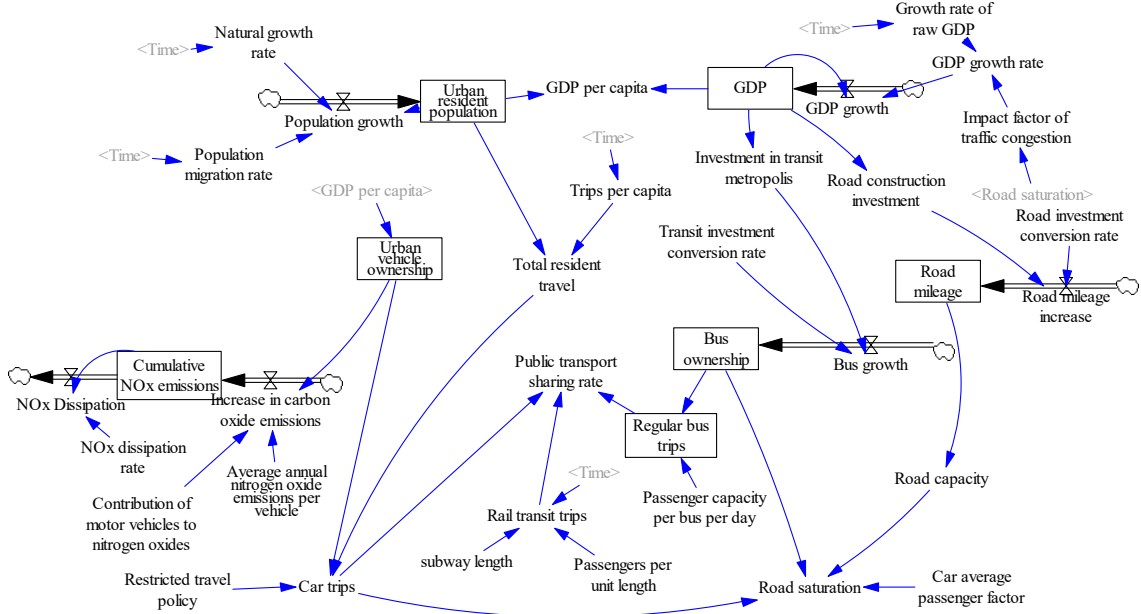

**Figure 3.** System flow diagram of transit metropolis construction.

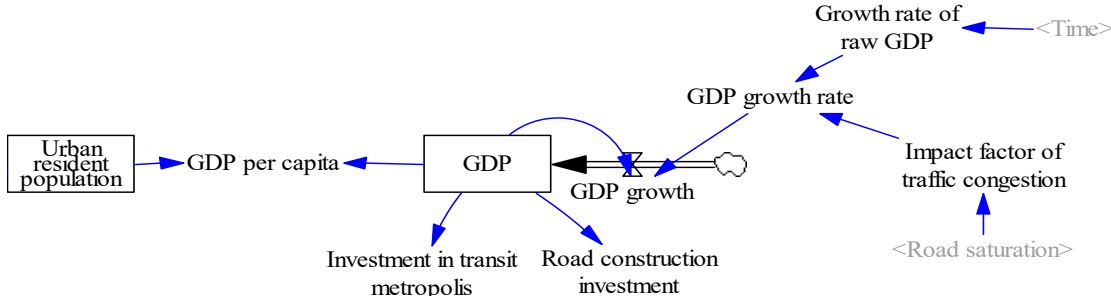

**Figure 4.** System causalities of the economic subsystem.

GDP is calculated from GDP points as follows, it equals the original GDP plus the increase GDP. This function is general for other cities. The specific data should be collected for a certain city.

$$R^{GDP} = \int_{t-1}^{t} GR^{GDP} \times OR^{GDP} dt + OR^{GDP} \tag{1}$$

where $R^{GDP}$ is the regional GDP, $GR^{GDP}$ is the GDP growth rate, $O^{GDP}$ is the original GDP. The GDP growth rate $GR^{GDP}$ is determined by the original GDP growth rate and the influence factor of traffic congestion, and the formula is as follows:

$$GR^{GDP} = OGR^{GDP} \times (1 + C^{TC}), \tag{2}$$

where $C^{TC}$ is the coefficient of traffic congestion. $OGR^{GDP}$ is the original GDP growth rate, and it is a table function related to time, and its expression is Function (3) as follows. $OGR^{GDP}$ is based on the *Nanchang Economic and Social Statistical Yearbook*. Therefore, if the SD model is used to describe other

cities, the data of the table function need to refer to the historical data of the local city. The data pair of the original GDP growth rate and the related year are shown in Figure 5.

$$OGR^{GDP} = \text{WITH LOOKUP (Time, } [(2008, 0) - (2025, 1)], (2008, 0.15), (2009, 0.141),$$
$$(2010, 0.15), \ (2011, 0.14), (2012, 0.135), (2013, 0.117), \ (2014, 0.108), \quad (3)$$
$$(2015, 0.106), (2016, 0.09), (2017, 0.09), \ (2025, 0.089))).$$

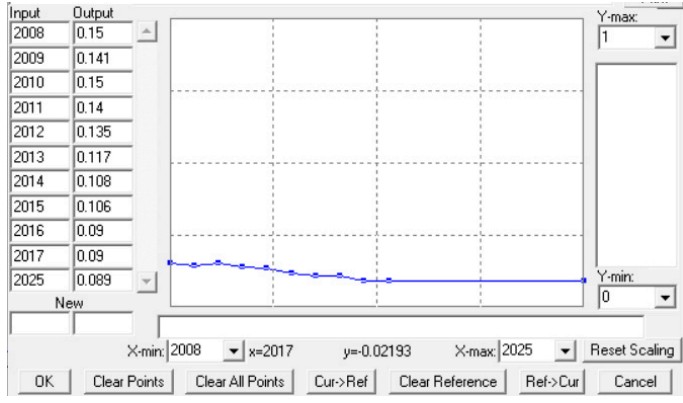

**Figure 5.** The data pair of the original GDP growth rate and the related year.

The influence factor of traffic congestion $C^{TC}$ is a table function of road saturation $R^S$, and its expression is as follows. The data pair of the coefficient of influence factor and road saturation are shown in Figure 6.

$$C^{TC} = \text{WITH LOOKUP}(R^S, [(0, -0.5) - (20, 0)], (0, 0), (0.5, -0.0012), (1, -0.0022),$$
$$(1.5, -0.003), (2, -0.0035), \ (2.5, -0.004), (3, -0.0045), (5, -0.005), (10, -0.01), (20, -0.02)). \quad (4)$$

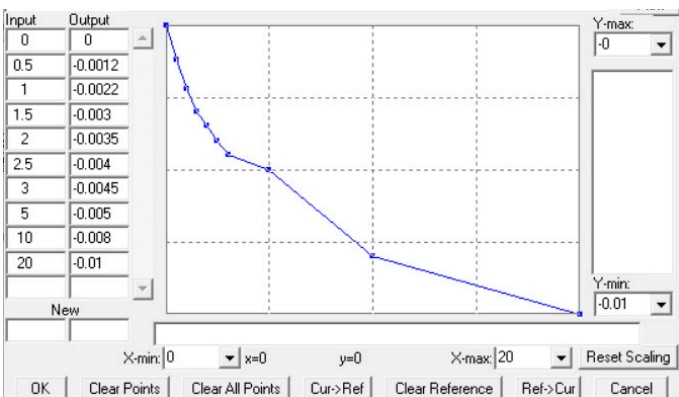

**Figure 6.** The data pair of the coefficient of influence factor and road saturation.

The GDP per capita $PC^{GDP}$ is divided by the regional GDP $R^{GDP}$ divided by the urban resident population $P^{UR}$. The formula is as follows:

$$PC^{GDP} = R^{GDP}/P^{UR} \quad (5)$$

The investment in transit metropolis $I^{TM}$ is determined by the product of regional GDP $R^{GDP}$ and public transport investment ratio $IR^{PT}$:

$$I^{TM} = R^{GDP} \times RI^{PT} \tag{6}$$

The investment in urban road construction $I^{URC}$ is determined by the product of the regional GDP $R^{GDP}$ and the proportion of urban road investment $P^{URI}$:

$$I^{URC} = R^{GDP} \times P^{URI} \tag{7}$$

4.3.2. Flow Chart and Dynamic Equation of the Society Subsystem

In the social subsystem (Figure 7), there is a level variable of the city's resident population, and the rate variables include the natural growth rate, population migration rate, and population growth. There are two table functions in the social subsystem, which are the table function of natural population growth rate and table function of population migration rate.

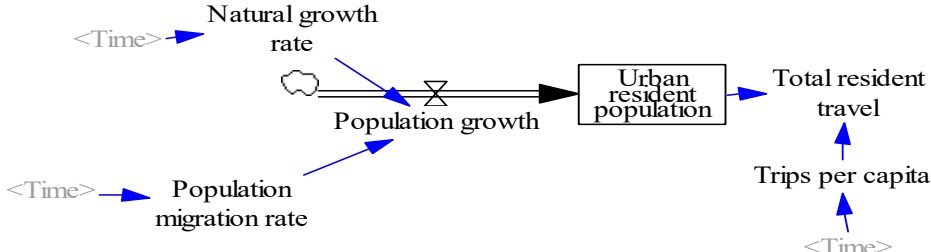

**Figure 7.** System causalities of the society subsystem.

The urban resident population $P^{UR}$ is calculated from the integral of population growth over time:

$$P^{UR} = \int_{t-1}^{t} (P^{GR} + 1) \times P^O dt \tag{8}$$

where $P^{GR}$ is population growth rate, $P^O$ is original population.

$$P^G = P^{UR} \times (N^{GR} + P^{MR}) \tag{9}$$

where, $P^{GR}$ is population growth, $N^{GR}$ is the natural growth rate of population, $P^{MR}$ is the population migration rate. The natural growth rate $N^{GR}$ is based on the data of the *National Economic and Social Development Bulletin of Nanchang* [37], which is a table function related to time. Therefore, the data of the table function need to refer to the historical data of the local city if the SD model is used to describe transit metropolis construction for other cities. The data pair of the natural growth rate of population and the related year are shown in Figure 8.

$$
\begin{aligned}
N^{GR} = \text{WITH LOOKUP } (&\text{Time}, [(2008, 0) - (2025, 0.03)], \ (2008, 0.00785), \ (2009, 0.00783), \\
&(2010, 0.0075), \ (2011, 0.0073), (2012, 0.007), \\
&(2013, 0.00679), (2014, 0.00678), (2015, 0.00675), (2016, 0.00693), \\
&(2017, 0.00763), (2018, 0.00683), (2025, 0.0065)).
\end{aligned}
\tag{10}
$$

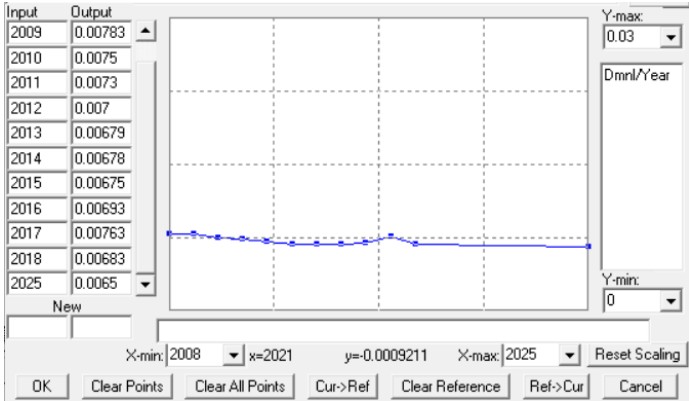

**Figure 8.** The data pair of the natural growth rate of population and the related year.

The population migration rate $P^{MR}$ is a table function related to time. It also needs local data for different cities. The data pair of the population migration rate and the related year are shown in Figure 9.

$$P^{MR} = \text{WITH LOOKUP (Time, } [(2008, 0) - (2025, 0.01)], \ (2008, 0.002),$$
$$(2012, 0.0035), (2013, 0.0038),$$
$$(2014, 0.00395), (2015, 0.00511), (2016, 0.00586), (2017, 0.0093), (2018, 0.008),$$
$$(2025, 0.009)).$$

(11)

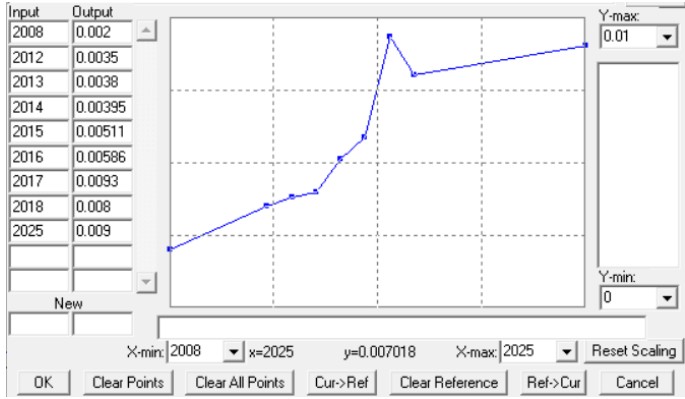

**Figure 9.** The data pair of the population migration rate and the related year.

The number of resident trips $N^{RT}$ is the product of the resident population $P^{UR}$ and the number of trips per resident $N^{PRT}$. The number of trips per capita $N^{PRT}$ is a table function related to time. It needs local data in the table function. The data pair of the number of trips per capita and the related year is shown are Figure 10.

$$N^{PRT} = \text{WITH LOOKUP (Time, } [(2008, 2) - (2024, 3)], (2008, 2.55), (2012, 2.6),$$
$$(2015, 2.65), \ (2018, 2.75), (2021, 2.85), (2023, 2.95)).$$

(12)

$$N^{RT} = P^{UR} \times N^{PRT}$$

(13)

where $N^{RT}$ is the total number of resident trips.

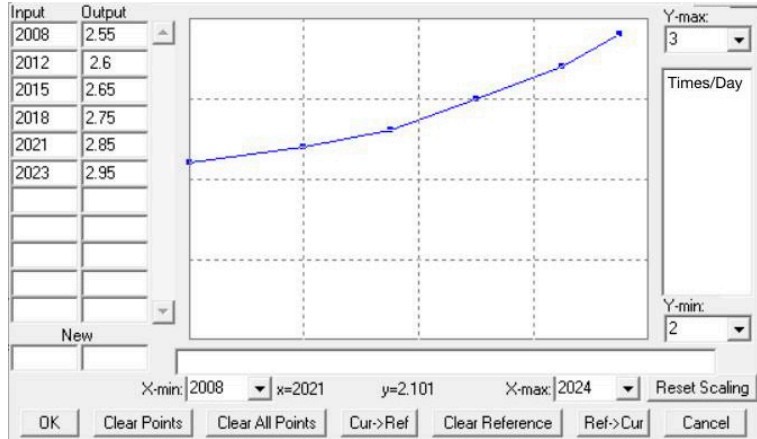

**Figure 10.** The data pair of the number of trips per capita and the related year.

### 4.3.3. Flow Chart and Dynamic Equation of the Environment Subsystem

The environmental subsystem (Figure 11) includes a level variable of cumulative nitrogen oxide emissions. The rate variables include nitrogen oxide dissipation, nitrogen oxide dissipation rate, nitrogen oxide emissions increase, motor vehicle contribution to nitrogen oxides, and vehicle average nitrogen oxide emissions. Urban vehicle ownership is a linear function related to per capita GDP, which is obtained from the economic system.

$$E^{NO_x} = \int_{t-1}^{t} (EI^{NO_x} - EI^{NO_x}) dt \tag{14}$$

where $E^{NO_x}$ is $NO_x$ emission, $EI^{NO_x}$ is emission increase of $NO_x$, $ED^{NO_x}$ is emission dissipation of $NO_x$.

$$ED^{NO_x} = EA^{NO_x} \times EDR^{NO_x} \tag{15}$$

where $EA^{NO_x}$ is $NO_x$ emission amount, $ED^{NO_x}$ is emission dissipation rate of $NO_x$.

$$EI^{NO_x} = UV^O \times EPV^{NO_x} \times MVC^{NO_x} \tag{16}$$

where $EI^{NO_x}$ is emission increase of $NO_x$, $UV^O$ is urban vehicle ownership, $EPV^{NO_x}$ is $NO_x$ emission per vehicle, $MVC^{NO_x}$ is motor vehicle contribution to $NO_x$ emissions. The number of motor vehicles will change with the residents' living standards. The number of motor vehicles is proportional to GDP per capita. This article uses historical data such as Nanchang's per capita GDP and Nanchang's motor vehicle ownership [37]. We performed a linear regression on the two to get the vehicle ownership equation as follows (Figure 12 and Formula 17). It needs the local data for the researched city.

$$UV^O = PC^{GDP} \times 10.64 - 22325 \tag{17}$$

where $PC^{GDP}$ is GDP per capital. In Equation (17), $R^2 = 0.938$**,** the regression data fits well and has statistical significance.

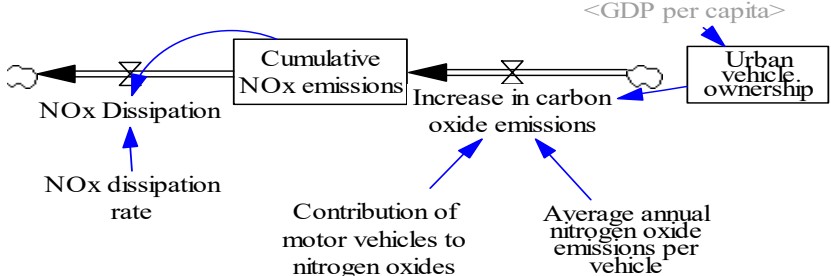

**Figure 11.** System causalities of the environment subsystem.

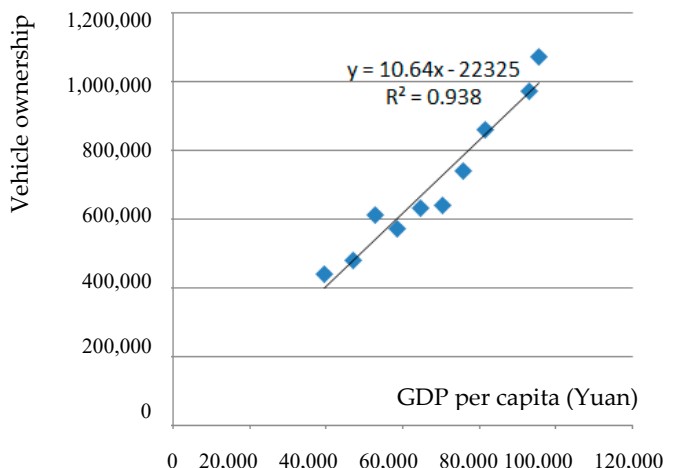

**Figure 12.** Relationship between GDP per capita and vehicle ownership.

#### 4.3.4. Flow Chart and Dynamic Equation of Transportation Demand and Supply

There are three level variables in the transportation supply and demand system (Figure 13), which are bus ownership, public transportation trips, and road mileage. The rate variables include bus growth and road mileage growth.

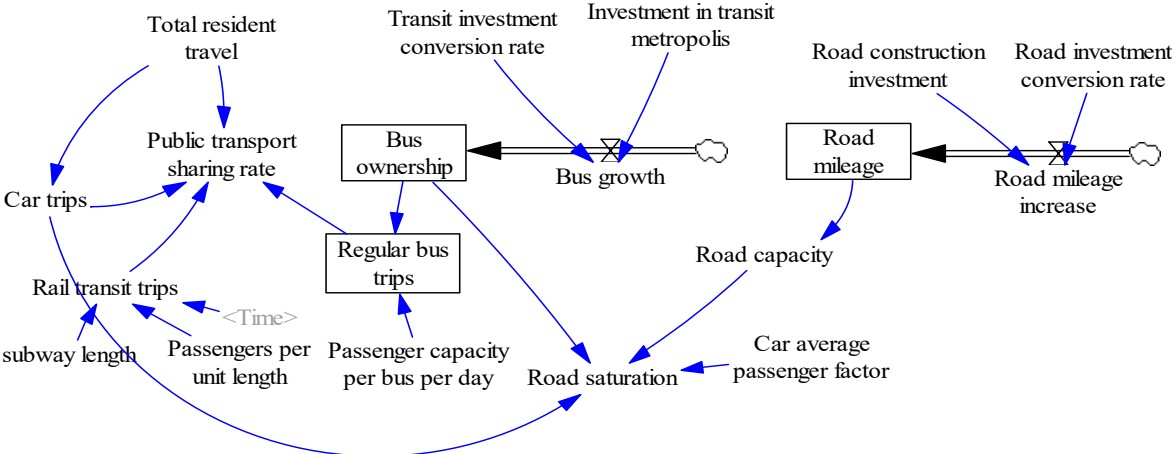

**Figure 13.** System causalities of the transportation demand and supply subsystem.

The amount of bus ownership is the integral of the growth of the bus over time:

$$B^O = \int_{t-1}^{t} B^{OI}dt + B^{OO} \tag{18}$$

where $B^O$ is bus ownership, $B^{OI}$ is bus ownership increase, $B^{OO}$ is original bus ownership.

$$B^{OI} = TI^{CR} \times I^{TM} = TI^{CR} \times R^{GDP} \times I^{TR} \times IR^{PT} \tag{19}$$

where $TI^{CR}$ is transit investment conversation rate, $I^{TM}$ is transit metropolis investment, $R^{GDP}$ is regional GDP, $I^{TR}$ is transportation investment ratio, $IR^{PT}$ is investment ratio of public transport.

Road mileage is the integral of road mileage increase over time:

$$R^M = \int_{t-1}^{t} R^{MI} dt + OR^M \tag{20}$$

where $R^M$ is road mileage, $R^{MI}$ is road mileage increase, $OR^M$ is original road mileage.

$$R^{MI} = I^{RC} \times RI^{CR} = R^{GDP} \times I^{TR} \times I^{RR} \times RI^{CR} \tag{21}$$

where $I^{RC}$ is road construction investment, $RI^{CR}$ is road investment conversion rate, $R^{GDP}$ is regional GDP, $I^{TR}$ is transportation investment ratio, $I^{RR}$ is road investment ratio.

Road capacity ($R^C$) calculation formula is as follows:

$$R^C = \left(R^M \times 0.05 \times 6 \times 15000\right) + \left(R^M \times 0.15 \times 6 \times 8000\right) + \left(R^M \times 0.8 \times 2 \times 5000\right) \tag{22}$$

The formula for calculating the number of rail transit trips is shown in Equation (23):

$$V^{RT} = S^L \times P^{PUSL} \tag{23}$$

where $V^{RT}$ is rail trip volume, $S^L$ is subway length, $P^{PUSL}$ is passengers per unit subway length.

$$RB^T = B^O \times PC^{PBPD} \tag{24}$$

where $RB^T$ is regular bus trips, $B^O$ is bus ownership, $PC^{PBPD}$ is passenger capacity per bus per day.

$$B^{MSR} = (RB^T + RT^T)/(RB^T + C^T + RT^T) \tag{25}$$

where $B^{MSR}$ is bus motorized sharing rate, $B^{MSR}$ is regular bus trips, $RT^T$ is rail transit trips, $C^T$ is car trips.

$$C^T = N^{RT} \times (1 - B^{MSR}) \tag{26}$$

where $N^{RT}$ is the total number of resident trips, $B^{MSR}$ is bus motorized sharing rate.

$$R^S = (C^T/C^{APF} \times 20 + B^O \times 200)/R^C \tag{27}$$

where $R^S$ is road saturation, $C^T$ is car trips, $C^{APF}$ is car average passenger factor, $B^O$ is bus ownership, $R^C$ is road capacity.

Based on the data provided by *Nanchang Statistical Yearbook* [37], *Statistical Bulletin*, and public transport collective, the initial values of the SD model parameters are given in Table 4. If the SD model is transferred to other cities for system analysis of transit metropolis construction, the initial data should be collected from the researched city.

**Table 4.** Initial values of parameters in the SD model.

| Parameter | Initial Value | Parameter | Initial Value |
|---|---|---|---|
| Regional GDP | RMB 16.60 billion | Urban resident population | 4.95 million |
| NOx emissions | 36 million tons | Contribution of motor vehicles to NOx [38] | 0.8 |
| NOx dissipation rate | 0.2 | Average annual NOx emissions per vehicle | 0.05 ton |
| Bus ownership of Nanchang | 2520 | Transit investment conversion rate | 1.5 |
| Proportion of transportation investment in regional GDP | 0.022 | Transit investment ratio in transportation investment | 0.25 |
| Road mileage | 1000 km | Road investment conversion rate | $1.71 \times 10^{-8}$ |
| Road investment ratio in transportation investment | 0.21 | Subway length | 48.47 km |
| Passenger per unit subway length | 6500 passenger/km | Passenger capacity per bus | 700 passengers/day |
| Car average travel distance | 20 km/day | Average passenger factor of car | 1.8 |
| Public transport travel distance | 250 km/day | | |

## 5. SD Model Simulation of Transit Metropolis Construction

### 5.1. Model Verification

No model precisely matches the real system because all models are something less than the system modelled [27]. Generally, SD model builders are more interested in dynamic tendencies than in specific values of system variables [27]. In practice, the usefulness of a proposed model is of major concern [39]. Forrester and Senge [40] believe that confidence is the best criterion for model assessment because there is no absolute proof of the model capability to describe reality. SD models are found to be valid when they can be used with confidence [39,40]. For SD model validation, it is common to compare the simulation values and actual data [22–32,38–41]. Sun [41] chose two variables (public transport vehicles and public transport demand) to test the SD model of the public transport price strategy. Eirini Grammatiki Pagoni [32] used historical data of Population_Net_Fractional_Rate, Income_Per_Capita, and Labour_Force_Participation to estimate parameters of the SD model and their trends. Xue [22] also chose two variables: public transport demand (trips/month) and public transport vehicles as test variables. If the reliability level is used as a test standard, the error rate of the SD model should be within 5% to meet the accuracy requirements [22,40,41]. When analyzing the parking charging policy in urban areas with a system dynamics model, He Jie [23] chose road length and vehicle ownership as the main historical value inspection indicators. There are three reasons for choosing these two parameters as test variables [23]: (1) the core of the research is the relationship between parking pricing policies and socioeconomic and urban transportation. The state variables in the model, road length and vehicle ownership, are the main variables; (2) These two variables have historical data that can be compared; (3) In the process of system dynamics simulation, according to the index interference level and the cyclic process, these variables are the center of the simulation process and are important factors affecting the change of the parking rate.

We will also use historical data to check the SD model validation according to the above SD literatures. Using the historical data of Nanchang, the two indicators of resident population and regional GDP of Nanchang were tested in Table 5. Additionally, the number of buses, mileage, and vehicle ownership can be used as indicators to verify the traffic facilities. The simulated values and actual data of the three parameters of Nanchang city in 2016 are compared in Table 6. From the data in Tables 5 and 6, it can be seen that from 2008 to 2018, the error rates of resident population and regional GDP of Nanchang City were controlled within 5%, the error rates of traffic facilities were also less than 5%, indicating that the accuracy of the model meets the needs of simulation analysis [22,40,41]. We can use the constructed system dynamics model to simulate and analyze the construction of Nanchang's transit metropolis.

**Table 5.** Comparison of historical and simulated data for the resident population (million) and GDP (billion).

| Year | Simulated Value of Resident Population | Real Value of Resident Population | Error Rate | Simulation Value of GDP | Real GDP | Error Rate |
|------|------|------|------|------|------|------|
| 2008 | 494.73 | 494.73 | 0.00% | 166.008 | 166.008 | 0.00% |
| 2009 | 499.60 | 497.33 | 0.45% | 190.825 | 183.750 | 3.71% |
| 2010 | 504.70 | 502.23 | 0.49% | 217.640 | 220.711 | 1.41% |
| 2011 | 509.88 | 504.95 | 0.98% | 250.173 | 268.887 | 7.48% |
| 2012 | 515.19 | 507.87 | 1.44% | 285.075 | 300.052 | 5.25% |
| 2013 | 520.60 | 518.42 | 0.42% | 323.424 | 338.726 | 4.73% |
| 2014 | 526.11 | 524.02 | 0.40% | 361.129 | 370.555 | 2.61% |
| 2015 | 531.76 | 530.29 | 0.28% | 399.991 | 401.188 | 0.30% |
| 2016 | 538.07 | 537.14 | 0.17% | 442.238 | 439.568 | 0.06% |
| 2017 | 544.95 | 546.35 | 0.26% | 481.895 | 481.976 | 0.02% |
| 2018 | 554.17 | 554.55 | 0.07% | 525.107 | 527.467 | 0.45% |

**Table 6.** Comparison of historical and simulated data for the traffic facilities of Nanchang city in 2016.

| Parameter | Simulated Value | Real Value | Error Rate |
|------|------|------|------|
| Number of buses (unit vehicle) | 3945 | 4000 | 1.37% |
| Mileage (Km) | 1181.6 | 1200 | 1.53% |
| Vehicle ownership (unit vehicle) | 852,181 | 860,000 | 0.91% |

### *5.2. Model Simulation and Analysis*

A complete analysis of the SD model's behavior requires the study of a large number of model parameters. Such a detailed experimental design is practically impossible and not necessary [32]. We will focus on the key factors related to the research topic. This section analyzes the SD model of transit metropolis construction, and analyzes the effects of Nanchang's transit metropolis construction with the main indicators of vehicle ownership, motorized travel-sharing rate of public transport, road saturation, and nitrogen oxide emissions. The simulation time of this model is from 2008 to 2023, and the simulation step is one year, which simulates the changing trend of Nanchang's urban transportation system. Because the academic edition of the software Vensim does not include sensitivity analysis function, we adjusted the values of the strategy parameters to obtain the value table of the specific variable via Vensim software. Then Excel was used to draw respective variable trend lines in the same figure under different strategies.

### 5.2.1. Vehicle Ownership

According to Vensim simulation results, the motor vehicle ownership rates of Nanchang City have increased linearly (shown in Figure 14). It is estimated that by 2020, the number of motor vehicles in Nanchang City will reach 1.13 million and the total urban population will reach 5.71 million (Table 4). According to this trend, the traffic problems in Nanchang will become more and more obvious. Therefore, it is necessary to carry out the construction of a transit metropolis.

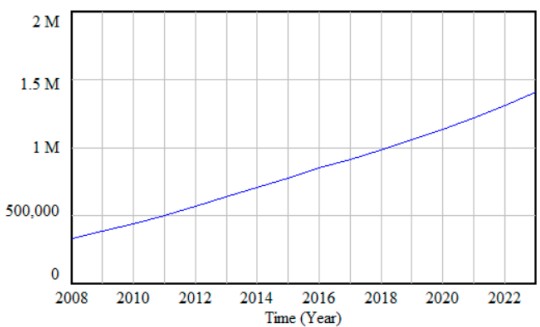

**Figure 14.** Motor vehicle ownership trends of Nanchang city.

### 5.2.2. Bus Motorized Travel Share Rate

It can be seen from Figure 15 that the share of public motorized travel in Nanchang is increasing year by year. In 2016, the share of public motorized travel in Nanchang increased from 49.7% to 52.7%. This is because Nanchang opened the first subway on 26 December 2015; citizens have more choices in public transportation. In 2018, Nanchang's bus motorized travel share rate was about 54%. Due to many factors such as road congestion and low bus speeds, citizens of Nanchang chose more individual travel; however, there is still a gap between the reality and the expected target of 60% for creating a transit metropolis if the original proportion of public transport investment is 0.25. Seen from Figure 15, through adjusting the proportion of public transport investment in the total urban investment, the motorized share rate of public transport can change respectively. Investment is critical for transit metropolis construction to realize the most important index—motorized share rate of public transport. This is because more investment in public transport can be used to construct more bus lanes and subways, buy more buses, maintain the ticket price at a low level. This strategy can attract more residents to travel by public transport.

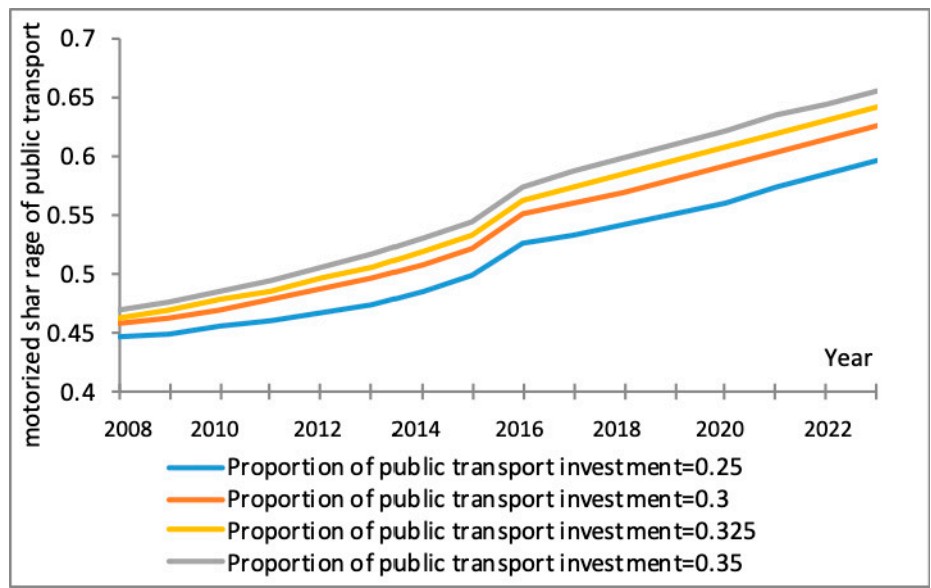

**Figure 15.** Trend of the share of public motorized travel in Nanchang.

### 5.2.3. NOx Emissions

Figure 16 shows the NOx emissions of motor vehicles in Nanchang from 2016 to 2022, and the values are calculated from the SD model. It can be seen that the NOx emissions of motor vehicles in Nanchang have been increasing year by year. The specific values are shown in Table 7:

According to the *China Motor Vehicle Environmental Management Annual Report* (2017), Jiangxi Province's total vehicle nitrogen oxide emissions in 2017 were 210,000 tons, and Nanchang's share exceeded that of Jiangxi Province's vehicle nitrogen oxide emissions by one-sixth. The number of motor vehicles in Nanchang continues to increase, and traffic pollution has become more serious. It is necessary to replace the present vehicles with new energy vehicles and decrease the annual vehicle exhaust emissions. In many large cities, such as Beijing and Shenzhen, the contribution rate of motor vehicle emissions has reached more than 40%, which is the culprit of environmental pollution, and it is urgent to reduce motor vehicle emissions.

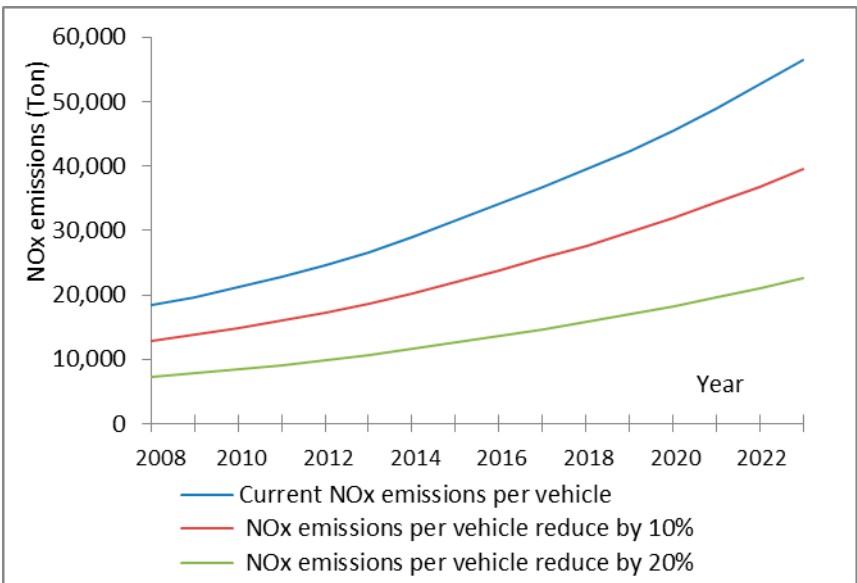

**Figure 16.** Trend of NOx emissions in Nanchang.

**Table 7.** Vehicle NOx Emissions in Nanchang.

| Year | 2016 | 2017 | 2018 | 2019 | 2020 | 2021 | 2022 |
|---|---|---|---|---|---|---|---|
| NOx emissions (tons) | 34.082 | 36.737 | 39.428 | 42.396 | 45.573 | 48.973 | 52.610 |

### 5.2.4. Traffic Congestion Factor (Road Saturation)

Figure 17 shows that the traffic congestion factor of Nanchang City is increasing year by year, and it is expected to reach a peak of 2.16 in 2022 if the original proportion of public transport investment is 0.25. This shows that Nanchang City has improved traffic demand and methods by improving road network grading and the construction of the transit metropolis, which has improved the traffic situation. According to Schrank's definition of the traffic congestion coefficient [42], when the traffic congestion coefficient is greater than 2, urban traffic is in a moderately congested state; when the traffic congestion coefficient is between 1 and 2, urban traffic is in a basically smooth state; when the traffic congestion coefficient is less than 1, urban traffic is in a smooth state. If the proportion of public transport investment is increased, the road saturation will be improved. It can be seen that during the future development of Nanchang, it will be in a state of moderate congestion, and the traffic situation in Nanchang will improve after 2022, indicating that the construction of the transit metropolis will have had positive effects on Nanchang's traffic.

### 5.3. Suggestions for the Transit Metropolis Construction of Nanchang City

Nanchang City needs to complete the requirements of the transit metropolis construction, and the share of public transport motorized travel needs to reach 60%. According to the simulation results, Nanchang City cannot achieve this goal in 2020. In the SD model, we adjusted the relevant indicators of Nanchang's transit metropolis construction to find a way to reach the share rate indicator by the end of 2020. The proportion of public transport investment in the total transportation investment was adjusted from 0.25 to 0.325; that is, about 900 additional buses were added to get a new share of public motor vehicle travel in Nanchang. According to Figure 15, at the end of 2020, Nanchang's public motor vehicle travel sharing rate will reach 61%, which is in line with Nanchang's transit metropolis construction index, and which indicates that Nanchang should increase its investment in public transport vehicles and infrastructures. It is still unable to meet the needs of residents' travel. It

is recommended to: (1) increase the number of buses, and (2) optimize the bus routes in Nanchang to reduce the transfer rate of public bus trips.

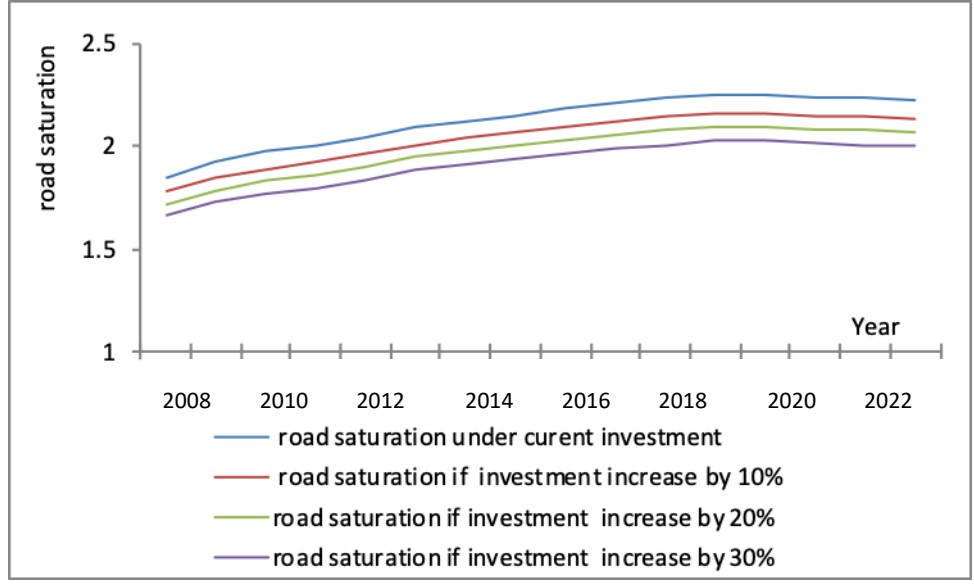

**Figure 17.** Trend of road saturation in Nanchang.

According to the simulation results of the above model, the vehicle emissions and pollutant emission speed of Nanchang City are increasing year by year. New energy vehicles are being promoted at home and abroad to reduce vehicle emissions. Moreover, the Transit Metropolis Creation Project clearly puts forward that the proportion of green public transport vehicles is an important assessment indicator. Therefore, through the construction of a transit metropolis, vehicle pollutant emissions will be improved. By adjusting the annual vehicle exhaust emissions from 0.05 tons to 0.04 tons per year a new cumulative nitrogen oxide emission in Nanchang can be obtained. Compared with the current simulation value, the nitrogen oxide emissions have decreased by nearly 70%, which indicates that the current automotive nitrogen oxide emissions are too high, and low-emission or new energy vehicles need to be developed. The creation of a transit metropolis will effectively improve the urban transportation environment.

## 6. Conclusions

Taking Nanchang's transit metropolis construction and its indicators as the starting point, the SD software Vensim is used to model some easily quantifiable indicators of Nanchang's transit metropolis. The core indicator of Nanchang's transit metropolis construction is the public transport motorization share rate, and nitrogen oxide emissions and traffic congestion coefficients were forecasted to illustrate the effect of Nanchang's transit metropolis construction. Due to the similarity of the indicators of the transit metropolis, the research results can be extended to other cities that struggle to become National Transit Metropolis Demonstration Cities.

Due to the rapid development of public transportation in Nanchang, the traffic data of Nanchang found on the Internet have a certain lag, and the SD model established this time only selects some of the core indicators of the construction of the transit metropolis, and the system indicators for modeling and evaluating need further improvement. In the future development process, due to the popularity of big data and intelligent transportation, many traffic data can be easily collected, which can also be updated and constructed more dynamically to improve the accuracy of the model.

Actually, the SD model has a delay phenomenon. Delay is a process whose output lags behind its input in some modes. For simplicity, this article ignores these delays and they need to be considered further in future research. Additionally, both the system dynamics and nonlinear specification of public

transportation supply and demand, as well as related causal relations and cost/evaluation methods need further and deeper research.

**Author Contributions:** The individual contributions and responsibilities of the authors are listed as follows: Y.X. designed the research, developed the model, conducted model validation, and wrote the paper; L.C. guided the research process; K.W. collected and analyzed the data; J.A. revised the manuscript, provided some comments and helped edit the manuscript; H.G. provided some comments on the case study and edited the manuscript. All authors have read and approved the final manuscript.

**Funding:** This research was sponsored by the National Natural Science Foundation of China (Grant No. 71961006 and 71971005), the Social Science Planning Fund of Jiangxi Province, China (No.18GL37), the College Humanities and Social Sciences Fund of Jiangxi Province, China (No. GL18219), and the Postdoctoral Research Foundation of Southeast University (No.1121000301).

**Acknowledgments:** The authors give thanks to the Nanchang Public Traffic Company for operational data. The authors are also very grateful for the comments from the editor and the anonymous reviewers.

**Conflicts of Interest:** The authors declare no conflict of interest.

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
