# Peer review of "System Dynamics Analysis of the Relationship between Transit Metropolis Construction and Sustainable Development of Urban Transportation—Case Study of Nanchang City, China"

_sustainability, doi:10.3390/su12073028_

Round 1

Reviewer 1 Report

The substantial work presents an important topic with potential interest to the reader. However, it is weak in presentation of its methodology, including hypothesis building, causal verification, and parameterisation, and thus reduces the validity of its findings. Its conclusions are not well founded.

Lines 57-74. (a) The concept of Transit Metropolis is critical to the paper. Although it is reviewed in Section 2.1, it must be further elaborated in terms of Motivation, Needs and Requirements, and Expected Benefits in the Introduction. (b) Further, the introduction should clarify whether this paper will make a contribution to the state of the art or whether it will focus on a case study in China; and should further clarify the objective; for instance, is it described in Lines 164-168?

Section 2.1 that presents the State of the Art (SAR) is not well organised. It oscillates between general findings and local interests without properly setting the worldwide state-of-the-art perspective. State of Practice could be added but this should be distinct from SAR. A proper clarification in the Introduction would also benefit this Section.

Lines 129-134 do not clearly justify the need for Systems Dynamics. In addition, they include false claims. For instance, generically SD is not precise and does not "predict".

Lines 135-136 are too general or incomplete and do not support the text.

Lines 136-159 are provided in a disorganised way in which no hierarchy or criticality is evident.

Lines 204-207. Causality and correlation are two different notions that are offered in a seemingly equal footing, and this needs to be clarified.

Lines 207-211, and Figure 2 and beyond. There is description but no analysis. For example, which loops are critical, which are stable, which are not. No delays are depicted (Fatal error).

Section 4.3: The functional relationships and Tables should be further explained, indicating what is arbitrary, what is identity and what is supported by models (Fatal error).

Section 5.1: Verification of population and GDP does not imply verification of the model (Fatal error).

Lines 350-353. These findings were already known from earlier discussion; the reasoning does not support the conclusion (Fatal error).

Lines 370-376. Important data. How do these relate to the findings?

Section 5, presenting the findings of the work should be much more extensive, including deep reasoning on functionalities, causalities and sensitivity analysis (Fatal error).

Lines 404-407. Repeat earlier introduced material.

Author Response

Sustainability

Manuscript number: 727982

                                                                Yunqiang Xue

                                                                Assistant Professor

                                                                East China Jiaotong University

                                                                Nanchang 330013, China

                                                                Mar.7th.2020

To Whom It May Concern,

I am writing to you regarding the modifications made to our paper (manuscript number 727982: “System dynamics analysis of the relationship between transit metropolis construction and sustainable development of urban transportation——Case study of Nanchang city, China”) in light of the reviewers’ comments.

All of the comments from the three anonymous reviewers were carefully studied and followed when revising the paper. The details about our response to the reviewers’ comments are summarized in the attachment at the end of this letter for your convenience.

We truly appreciate the reviewers’ constructive comments and believe that the revised paper is now clearer and more useful to researchers and practitioners. If you have further questions, please feel free to contact me at [email protected]

Sincerely yours,

Yunqiang Xue

Attachment: Authors’ Response to Reviewers

Dear Reviewers,

Thank you so much for your time and valuable suggestions. The authors have carefully and seriously considered the questions and suggestions enclosed in the review and made necessary revisions.

The reviewers’ comments are in BOLD. The newly-added content in the article is in Blue. The deleted content in the article is in Pink. Other contents in the article are in italic type. Please see the attachment, the revised contents with color .

Reviewer 1:

Question 1:  The substantial work presents an important topic with potential interest to the reader. However, it is weak in presentation of its methodology, including hypothesis building, causal verification, and parameterization, and thus reduces the validity of its findings. Its conclusions are not well founded.

Response 1-1:

Thank you so much for the comment. . We have revised the manuscript according three anonymous reviewers. The revised manuscript has been resubmitted. We hope the revised paper is now clearer and more useful to researchers and practitioners. Please feel free to contact me if you have any questions. Thanks.

Question 2: Lines 57-74. (a) The concept of Transit Metropolis is critical to the paper. Although it is reviewed in Section 2.1, it must be further elaborated in terms of Motivation, Needs and Requirements, and Expected Benefits in the Introduction. (b) Further, the introduction should clarify whether this paper will make a contribution to the state of the art or whether it will focus on a case study in China; and should further clarify the objective; for instance, is it described in Lines 164-168?

Response 1-2-(a):

According to the comment, we have added motivation, needs and requirements, and Expected Benefits of Transit Metropolis in lines 59-89.

Response 1-2-(b):

Thank you so much for the comment. We have added descriptions in the last second paragraph of section 2(Lines231-245).

Question 3: Section 2.1 that presents the State of the Art (SAR) is not well organized. It oscillates between general findings and local interests without properly setting the worldwide state-of-the-art perspective. State of Practice could be added but this should be distinct from SAR. A proper clarification in the Introduction would also benefit this Section.

Response 1-3:

Thank you very much for your comments. According to the comments, we have added descriptions in the second and the fourth paragraph in section 2.1. The revised paragraphs are listed as follows:

….Although there are differences in transit metropolis and measurement indicators at home and abroad, there is a consensus that transit metropolis embody a structure of urban layout that uses urban public transportation as the main body of motorized travel and guides urban development with urban public transportation. Constrained by safety, resources, environment and other conditions, the best form of urban construction is an urban development model with the best overall efficiency and social and environmental benefits. Transit metropolis advocates urban public transportation to actively guide urban development, emphasizing that urban public transportation and urban human settlements, structural functions, the environment, and spatial layout are coexist, harmonious and promote. Transit metropolis is a leap in understanding the theory and practice to solve urban and transportation problems.

Nowadays, China is the only country in the world which adopts a top-down program to actively promote transit metropolis nationwide [14]. The related researches and ongoing transit metropolis efforts in China would help to generate more benefits for sustainable transportation and promote more transit metropolises in the future. China's transit metropolis construction experience will provide reference for other cities in the world to develop public transit and promote urban sustainable transportation development.

Question 4: Lines 129-134 do not clearly justify the need for Systems Dynamics. In addition, they include false claims. For instance, generically SD is not precise and does not "predict".

Response 1-4:

Thank you very much for your comments. According to the comments, the first paragraph of section 2.2 is revised as follows:

The construction of transit metropolis is a systematic project, and the public transport system itself is also a complex system with multiple feedbacks, variables, and nonlinearities [22]. The primary motivation for adopting System Dynamics (SD) as the methodology is that the research needs to represent concurrent and multiple intersections among variables in different subsystems. The SD approach can allow one to understand and interpret the interactions easily [22]. Furthermore, a key strength of the SD approach is that it could describe the dynamic processes which evolve continuously and with lags or time delays. This ability is important since we need to study the cumulative impacts of transit metropolis construction over years. Finally, the SD model can describe nonlinear relationships among variables [22]. The nonlinear interactions of multiple factors that are parts of decision-making, and the basic physics of systems will be potentially ignored if the nonlinear relationships are not considered. The SD approach can suitably describe the cumulative impacts and developing trends of transit metropolis construction. What is more, System Dynamics (SD)SD can precisely reflect the variable structure of the above complex system of transit metropolis construction, and can predict the trend and cumulative effect of system variables over time. It is obvious to research the transit metropolis program as a whole requires a systemic analysis approach. Therefore, it is applicable and feasible to use SD method to study the traffic problems such as the construction of transit metropolis, which can reflect the advantages of systemicity and dynamics.

(Note: The deleted content in the article is in Pink.)

Question 5: Lines 135-136 are too general or incomplete and do not support the text.

Response 1-5:

Thank you very much for your comments.

In fact, the first sentence of the second paragraph of Section 2.2 is a summary of the study of traffic problems using the SD method. The rest of this paragraph is summarized around the aspects listed in the sentence. In order to make the statement coherent, we have added some connection sentences in this paragraph. The detailed revision can be seen in the response to question 6.

Question 6: Lines 136-159 are provided in a disorganized way in which no hierarchy or criticality is evident.

Response 1-6:

Thank you very much for your comments. According to the comments we have added some connection sentences in this paragraph to make the statement coherent. The detailed revision is listed as follows:

The construction project of transit metropolis has Chinese characteristics. There are many researches on transit metropolis in China, but there is little research on the systematic effects of transit metropolis program. Many researchers used SD approach to analyze transportation system problems. Wilsses et al. [23] considered that it is applicable to capture the causality of urban transportation system variables using system dynamics, and analyzed the impact of some Brazilian policies on the transportation system environment, economy and transportation variables. Yongtao Tan et al.[24]used Beijing, China as an example to simulate the sustainable performance of the city with a system dynamics model. In order to achieve sustainable development, Beijing should adopt a low speed urbanization policy. SD approach is also used to research highway problems. Gokhan et al. [25] used a system dynamics method to study highway sustainability and simulated three potential strategies for policy development: fuel efficiency, public transportation, and electric vehicle usage. The mixed implementation of strategies plays an important role in the success of policy making. Additionally, SD method is used to study road traffic emission policies. Aiga Barisa et al. [26] used a system dynamics model to analyze CO2 emission reduction policies of road traffic. The SD model can better understand the factors behind road emissions. Xue Liu [27]proposed a SD approach to scenario analysis for energy consumption and CO2 emissions of urban passenger transport, and built a Beijing urban passenger transport carbon model. What is more, some authors used SD to analyze congestion policies. Shiyong Liu[28] used the SD model to assess the dynamic impact of congestion pricing strategies on the socio-economic system of transportation and to support policy makers. For the freight transportation problems, Carina Thaller et al.[29]used SD model to describe and reveal urban freight traffic, and made medium and long-term predictions. Wen Huang [30]proposed a methodological framework for truck weight regulation based on System Dynamics. The framework is composed of five subsystems and can capture the vehicle, highway and freight variables which influence the effects of truck weight regulation and transportation efficiency over time. For some other problems, Eirini Grammatiki Pagoni [31] developed a novel SD-based decision support tool to assess socially and financially sustainable performance of national Public-Private Partnership programmesprograms. The construction project of transit metropolis has Chinese characteristics. Although there are many researches on transit metropolis in China [10, 14-15, 17-21], there is little research on the systematic effects of transit metropolis program. 

(Note: The deleted content in the article is in Pink.)

Question 7: Lines 204-207. Causality and correlation are two different notions that are offered in a seemingly equal footing, and this needs to be clarified.

Response 1-7:

Thank you very much for your comments. According to the comments, we added explanations for causality diagram as follows:

After determining the scope and boundary of the model, the interactions between the four main boundaries of transportation supply and demand, environment, economy, and population are analyzed. Establish a causality diagram for the construction of a transit metropolis, as shown in Figure 2. The causality diagram is also called a system cycle diagram. Starting from the simple causality relationship between system elements, a relationship chain and a causality circuit are established to form a diagram of mutual constraints between the elements [22]. The interrelationships between a series of elements constitute a causal chain or a causal circuit. The plus and minus signs next to the arrows indicate the positive and negative attributes of the correlation between the variables. For example, the improvement of regional GDP has a positive effect on the investment of transit metropolis construction. The investment of transit metropolis construction has a positive correlation with the share of public transport, and has a positive correlation with the amount of public bus trips. The increase in the average speed of the road network is conducive to the improvement of road traffic conditions, and the degree of road congestion has an adverse effect on regional local GDP.

Question 8: Lines 207-211, and Figure 2 and beyond. There is description but no analysis. For example, which loops are critical, which are stable, which are not. No delays are depicted (Fatal error).

Response 1-8:

Thank you very much for your comments. A paragraph has been added at the end of section 4.2 as follows:

Causality diagrams describe the interrelationships between variables, and no quantitative analysis has yet been given. In the causality diagram stage, each cycle is important, and the key elements will be established after a system flow diagram is constructed for quantitative analysis. The quantitative relationship between the variables will be analyzed in the next section, discussing how a change in a variable affects the change in the system, and then determining which factor is the key factor affecting the system.

Question 9: Section 4.3: The functional relationships and Tables should be further explained, indicating what is arbitrary, what is identity and what is supported by models (Fatal error).

Response 1-9:

Thank you very much for your comments. We have revised all the formulas (1)-(27) in the revised manuscript and added more discussions. Please see the resubmitted manuscript.

Question 10: Section 5.1: Verification of population and GDP does not imply verification of the model (Fatal error).

Response 1-10:

Thank you very much for your comments. We have added the test criteria for the SD model to the first paragraph of section 5.1 as follows:

Sun [35] chose two variables (public transport vehicles and public transport demand) to test the SD model of the public transport price strategy.  Xue [22] also chose two variables public transport demand (trip/month) and public transport vehicles as test variables. If the reliability level is used as a test standard, the error rate of SD model should be within 5% to meet the accuracy requirements [22,35-36]. Using the historical data of Nanchang, the two indicators of resident population and regional GDP of Nanchang were tested. From the data in Table 4, it can be seen that from 2008 to 2018, the error rate of resident population and regional GDP of Nanchang City were controlled within 5% 49, indicating that the accuracy of the model meets the needs of simulation analysis [3522,35-36]. We can use the constructed system dynamics model to simulate and analyze the construction of Nanchang's transit metropolis.

Question 11: Lines 350-353. These findings were already known from earlier discussion; the reasoning does not support the conclusion (Fatal error).

Response 1-11:

Thank you very much for your comments.

Because resident population and GDP have been discussed for model verification, the tile of section 5.2.1 has been changed from “Vehicle ownership and urban resident population” to “Vehicle ownership”. The urban resident population trends of Nanchang city in figure 8 is also deleted.

The revised paragraph in section 5.2.1 is revised as follows:

According to Vensim simulation results, the resident population and motor vehicle ownership rate of Nanchang City have increased linearly. It is estimated that by 2020, the number of motor vehicles in Nanchang City will reach 1.13 million and the total urban population will reach 5.71 million (table 4). According to this trend, the traffic problems in Nanchang will become more and more obvious. Therefore, it is necessary to carry out the construction of a transit metropolis.

Question 12: Lines 370-376. Important data. How do these relate to the findings? reasoning does not support the conclusion (Fatal error).

Response 1-12:

Thank you very much for your comments.

Table 5 shows the vehicle NOx emissions in Nanching, and the values are calculated from SD model. Compared with total vehicle nitrogen oxide emissions of Jianxi Province in 2017, we can see Nanchang’s vehicle emissions share one-sixth of the total emissions. Nanchang’s vehicle emissions will be more and more serious, it is necessary to replace the present vehicles with new energy vehicles and decrease the annual vehicle exhaust emissions. The revised contents are listed as follows:

Figure 9(b) shows the NOx emissions of motor vehicles in Nanchang from 2016 to 2022, and the values are calculated from SD model. It can be seen that the NOx emissions of motor vehicles in Nanchang have been increasing year by year. The specific values are shown in the table below:

Table5.  Vehicle NOx Emissions in Nanchang

year

2016

2017

2018

2019

2020

2021

2022

NOx emissions(tons)

34 082

36 737

39 428

42 396

45 573

48 973

52 610

According to the China Motor Vehicle Environmental Management Annual Report (2017), Jiangxi Province’s total vehicle nitrogen oxide emissions in 2017 were 210,000 tons, and Nanchang ’s share exceeded that of Jiangxi Province ’s vehicle nitrogen oxide emissions by one-sixth. The number of motor vehicles in Nanchang has continued to increase, and traffic pollution has become more serious. It is necessary to replace the present vehicles with new energy vehicles and decrease the annual vehicle exhaust emissions. In many large cities, such as Beijing and Shenzhen, the contribution rate of motor vehicle emissions has reached more than 40%, which has become the culprit of environmental pollution, and it is urgent to reduce motor vehicle emissions.

Question 13: Section 5, presenting the findings of the work should be much more extensive, including deep reasoning on functionalities, causalities and sensitivity analysis (Fatal error).

Response 1-13:

Thank you very much for your comments. We have revised all the formulas and revised the section 5.2 model simulation. The figures and table in the section show the quantitative results. We also explained the symbol in the first paragraph of section section4.3.

According to the cause-effect diagram, the system flow diagram (Figure 3) for the construction of a transit metropolis is divided into four subsystems: economy, society, environment, and transportation supply and demand. The system flow diagrams and dynamic equations of these four subsystems are given below. The symbol means a rate variable, cloud     

represent source, the starting point of the rate variable. The black arrow points to a state variable. For example, population growth is a rate variable, the source of population is the initial value and it will change with the time, at the same time it points to a variable urban resident population.

Question 14: Lines 404-407. Repeat earlier introduced material.

Response 1-14:

Thank you very much for your comments. We have added some descriptions about transit metropolis and transport environment in the last paragraph of section 5.3. The revised paragraph is listed as follows:

According to the simulation results of the above model, the vehicle emissions and pollutant emission speed of Nanchang City have been increasing year by year. New energy vehicles are being promoted at home and abroad to reduce vehicle emissions. Moreover, the transit metropolis creation project clearly puts forward that the proportion of green public transport vehicles is an important assessment indicator. Therefore, through the construction of transit metropolis, vehicle pollutant emissions will be improved. By adjusting the annual vehicle exhaust emissions, if it is adjusted from 0.05 tons to 0.04 tons per year to obtain a new cumulative nitrogen oxide emission in Nanchang. Compared with the current simulation value, the nitrogen oxide emissions have decreased by nearly 70%, which indicates that the current automotive nitrogen oxide emissions are too high, and low-emission or new energy vehicles need to be developed. The creation of a transit metropolis will effectively improve the urban transportation environment.

Reviewer 2 Report

1. Background (the introduction) should present the current state of knowledge in the given subject. This article does not include a information about BRT systems (bus rapid transit, Curitiba - Brazil). 

2. No explanation of the symbol: Figure 3, Figure 4, Figure 5 - see Annex

3. An illegible mathematical formula (1) - see Annex

4. I propose that you change the wording of formulas eg.

a) formule (23) Rail trip volume = subway length × passengers per unit subway length;

RTV = SL x PSL,

where:

RTV - Rail trip volume, ...

b) formule (24) Regular bus trips = bus ownership x passenger capacity per bus per day 

RBT = BO x PCBD

and so on. 

5. The style of reference list is not in conformity with the requirements of MDPI regulations (see: https://www.mdpi.com/authors/references): eg.:

  • [2]. Miller, P., et al., Public Transportation and Sustainability: A Review. KSCE Journal of Civil Engineering, 2016. 20(3): p. 1076-1083.
  • [3]. Ghorbanzadeh, O., et al., Sustainable urban transport planning considering different stakeholder groups by an interval-AHP decision support model. Sustainability, 2018. 2018(12): p. 1-18. 

Author Response

Sustainability

Manuscript number: 727982

                                                                Yunqiang Xue

                                                                Assistant Professor

                                                                East China Jiaotong University

                                                                Nanchang 330013, China

                                                                Mar.7th.2020

To Whom It May Concern,

I am writing to you regarding the modifications made to our paper (manuscript number 727982: “System dynamics analysis of the relationship between transit metropolis construction and sustainable development of urban transportation——Case study of Nanchang city, China”) in light of the reviewers’ comments.

All of the comments from the three anonymous reviewers were carefully studied and followed when revising the paper. The details about our response to the reviewers’ comments are summarized in the attachment at the end of this letter for your convenience.

We truly appreciate the reviewers’ constructive comments and believe that the revised paper is now clearer and more useful to researchers and practitioners. If you have further questions, please feel free to contact me at [email protected]

Sincerely yours,

Yunqiang Xue

Attachment: Authors’ Response to Reviewers

Dear Reviewer,

Thank you so much for your time and valuable suggestions. The authors have carefully and seriously considered the questions and suggestions enclosed in the review and made necessary revisions.

The reviewers’ comments are in BOLD. The newly-added content in the article is in Blue. The deleted content in the article is in Pink. Other contents in the article are in italic type.Please see the attachment, the revised contents with color .

Reviewer 2:

Question 1: Background (the introduction) should present the current state of knowledge in the given subject. This article does not include information about BRT systems (bus rapid transit, Curitiba - Brazil).

Response 2-1:

Thanks for the reviewer’s comments. We have added the information about BRT systems at the end of the third paragraph in Section 1(Introduction). The third paragraph of section 1 is revised as follows:

Up to now, a total of three batches of 87 cities have been selected as Transit Metropolis creation demonstration projects. 14 cities, including Shanghai and Nanjing, have completed the establishment of Transit Metropolis, and have officially become "national transit metropolis demonstration cities." The application city for the “Transit Metropolis “program needs to refer to the requirements of 20 assessment indicators (Table 1) and 10 reference indicators (Table 2) [10], and propose the specific quantitative and qualitative goals of its own city. A city formally won the honorary title of "National Transit Metropolis Demonstration City" only after at least 5 years of creation and successful completion of the creation goals proposed in the application. In addition, the city applying for transit metropolis can also propose no more than 3 characteristic indicators based on its own characteristics. For example, due to the abundant underground spring water, Jinan City lacks large-capacity subway lines; the BRT (Bus Rapid Transit) network forms the backbone of the public transportation network. Therefore, Jinan takes the ratio of the length of the BRT line network to the length of the entire public transportation line network as a characteristic indicator of the creation of a transit metropolis. In fact, Curitiba, Brazil is the hometown of BRT. Many cities in the world, including most large and medium-sized cities in China, have planned and built BRT under the guidance of the Brazilian experts from the World Bank. The BRT system also plays an important role in the construction of transit metropolis. Robert Cervero [11-12] detailed the efficient BRT system in Curitiba, Brazil in his book "Transit Metropolis".

(Note: The newly-added content in the article is in Blue.)

Question 2: No explanation of the symbol: Figure 3, Figure 4, Figure 5 - see Annex

Response 2-2:

Thanks for the reviewer’s comments. We have explained the symbol in the first paragraph of section section4.3. The revised paragraph is listed as follows:

According to the cause-effect diagram, the system flow diagram (Figure 3) for the construction of a transit metropolis is divided into four subsystems: economy, society, environment, and transportation supply and demand. The system flow diagrams and dynamic equations of these four subsystems are given below. The symbol means a rate variable, cloud     

represent source, the starting point of the rate variable. The black arrow points to a state variable. For example, population growth is a rate variable, the source of population is the initial value and it will change with the time, at the same time it points to a variable urban resident population.

Question 3: An illegible mathematical formula (1) - see Annex

Response 2-3:

Thanks for the reviewer’s comments. We have revised formula (1) as follows:

                                     (1)

  Where, is the regional GDP, is the GDP growth rate, is the original GDP.The GDP growth rate  is determined by the original GDP growth rate and the influence factor of traffic congestion,

Question 4: I propose that you change the wording of formulas eg.

a) formule (23) Rail trip volume = subway length × passengers per unit subway length;

RTV = SL x PSL, where: RTV - Rail trip volume, ...

b) formule (24) Regular bus trips = bus ownership x passenger capacity per bus per day

RBT = BO x PCBD

and so on. An illegible mathematical formula (1) - see Annex

Response 2-4:

Thank you very much for the professional comment. We have revised all the formulas (1)-(27) in the revised manuscript. Please see the resubmitted manuscript.

Question 5: The style of reference list is not in conformity with the requirements of MDPI regulations (see: https://www.mdpi.com/authors/references): eg.:

•           [2]. Miller, P., et al., Public Transportation and Sustainability: A Review. KSCE Journal of Civil Engineering, 2016. 20(3): p. 1076-1083.

•           [3]. Ghorbanzadeh, O., et al., Sustainable urban transport planning considering different stakeholder groups by an interval-AHP decision support model. Sustainability, 2018. 2018(12): p. 1-18.

Response 2-5:

Thank you for the comment. We have revised reference list according to the journal requirements. Please see the revised manuscript. Thanks.

Reviewer 3 Report

I find the article both interesting and well written. A comprehensive exercise has been done to present a good manuscript with comprehensive effort. Scope and results are presented on satisfactory level. The used research approach is balanced; the proposed methodology is examined in case study in Nanchang city as well in Transit Metropolis condition. Referencing is done according instructions, there are cited many relevant authors. The paper represents a contribution to the sustainable public transport.

Author Response

Sustainability

Manuscript number: 727982

                                                                   Yunqiang Xue

                                                                   Assistant Professor

                                                                   East China Jiaotong University

                                                                   Nanchang 330013, China

                                                                   Mar.7th.2020

To Whom It May Concern,

I am writing to you regarding the modifications made to our paper (manuscript number 727982: “System dynamics analysis of the relationship between transit metropolis construction and sustainable development of urban transportation——Case study of Nanchang city, China”) in light of the reviewers’ comments.

All of the comments from the three anonymous reviewers were carefully studied and followed when revising the paper. The details about our response to the reviewers’ comments are summarized in the attachment at the end of this letter for your convenience.

We truly appreciate the reviewers’ constructive comments and believe that the revised paper is now clearer and more useful to researchers and practitioners. If you have further questions, please feel free to contact me at [email protected]

Sincerely yours,

Yunqiang Xue

Attachment: Authors’ Response to Reviewers

Dear Reviewer,

Thank you so much for your time and valuable suggestions. The authors have carefully and seriously considered the questions and suggestions enclosed in the review and made necessary revisions.

The reviewers’ comments are in BOLD. The newly-added content in the article is in Blue. The deleted content in the article is in Pink. Other contents in the article are in italic type.Please see the attachment, the revised contents with color .

Reviewer 3:

Question 1: I find the article both interesting and well written. A comprehensive exercise has been done to present a good manuscript with comprehensive effort. Scope and results are presented on satisfactory level. The used research approach is balanced; the proposed methodology is examined in case study in Nanchang city as well in Transit Metropolis condition. Referencing is done according instructions, there are cited many relevant authors. The paper represents a contribution to the sustainable public transport.

Response 3-1:

Thank you very much for the encouragement. We have revised the manuscript according three anonymous reviewers. The revised manuscript has been resubmitted. We hope the revised paper is now clearer and more useful to researchers and practitioners. Please feel free to contact me if you have any questions. Thanks.

Round 2

Reviewer 1 Report

The substantial work presents an important topic with potential interest to the reader. However, it is weak in presentation of its methodology, including hypothesis building, causal verification, and parameterisation, and thus reduces the validity of its findings. Its conclusions are not well founded. Further to the earlier evaluation, certain elements are selected for brevity.

The concept of Transit Metropolis (Section 2.1) should be elaborated in terms of Needs and Requirements, Expected Benefits and Transferability. The response by the authors indicates that, "China's transit metropolis construction experience will provide reference for other cities in the world to develop public transit and promote urban sustainable transportation development."  However, the presumed transferability properties cannot be defended from this manuscript as the local quality of performance has not been achieved.

Lines 129-134 do not clearly justify the need for Systems Dynamics. In addition, they include false claims. For instance, generically SD is not precise and does not "predict". The response by the authors has not repaired this deficiency.

Lines 136-159 are provided in a disorganised way in which no hierarchy or criticality is evident. Despite the useful addition by the authors, the position of this work within the SAR still does not clearly indicate the research linkages with the authors' work.

Lines 204-207. Causality and correlation are two different notions that are offered in a seemingly equal footing, and this needs to be clarified. The response by the authors does not clearly address this concern.

Lines 207-211, and Figure 2 and beyond. There is description but no qualitative analysis. For example, which loops are critical, which are expected to be stable, which not. This type of qualitative hypothesis should precede any subsequent quantitative analysis (Fatal error). Has not been addressed by the authors.

Section 4.3: The functional relationships and Tables should be further explained, indicating what is arbitrary, what is identity and what is supported by models (Fatal error). Has not been addressed by the authors.

Section 5.1: Verification of population and GDP does not imply verification of the model (Fatal error). Population is an overpowering variable over the rest and cannot offer a criterion for other variables. Has not been addressed by the authors.

Section 5, presenting the findings of the work should be much more extensive, including deep reasoning on functionalities, causalities and sensitivity analysis (Fatal error). Has not been addressed by the authors.

Round 3

Reviewer 1 Report

With respect to the earlier versions, the quality of presentation has increased. Nevertheless, the scientific soundness has not increased. For instance, at the nucleus of the proposed work, there exists an extensive body of literature (a short excerpt follows at the end if it could assist the authors) that has firmly quantified on scientific grounds, even supporting a Nobel Prize on certain grounds, both the systems dynamics/ nonlinear specification of public transportation demand & supply, as well as related causal relations and cost/evaluation methods. Further, anchoring public transport error on the characteristics of dominant variables, such as population must be avoided.

  1. Loudon, W.R. (1982). Relating performance measures to organizational objectives. MIT & Cambridge Systematics, prepared for TRB 61st ann mtg.
  2. Adler, T.J., S.R. Stearns and Y.J. Stephanedes (1980). "Techniques for Analyzing the Performance of Rural Transit Systems", Final Report, DOT-RSPA-DPB-50/80/23, U.S. Dept. of Transportation, Office of University Research, Washington, DC.
  3. Institute for Urban Transportation (1980). Mass transit management: A handbook for small cities. Indiana University, Bloomington, Indiana, September.
  4. Meadows, D.H. (1999). Leverage points: places to intervene in a system. The sustainability Institute. Hartland Four Corners, Vermont. Available at: http://www.sustainer.org [Oct 2001].
  5. Stephanedes Y.J. (1982). Control applications in analyzing transportation system performance under dynamic constraints. In: Drenick & Kozin (eds) System Modeling and Optimization. Lecture Notes in Control Information Sciences, vol.38. Springer, Berlin.
  6. Stave, K.A. (2002). Using system dynamics to improve public participation in environmental decisions Syst. Dyn. Rev. 18, 139–167.
  7. Carter-Goble Associates (1982). Rural public transportation performance evaluation guide. Final Report, Pennsylvania Department of Transportation, Harrisburg, Pennsylvania. Technology Sharing Program Off. Secretary of Transportation, Washington, DC.
  8. McFadden, D. (1973). "Conditional Logit Analysis of Qualitative Choice Behavior", Zarembka, P., ed., Frontiers in Econometrics, Academic Press, New York, NY, 1973.
  9. Stephanedes, Y.J, Adler, T. (1979). Forecasting experiments for rural transit. TRB 58th ann. mtg.
  10. Ben-Akiva, M. and T. Atherton (1976). "Transferability and Updating of Disaggregate Travel Demand Models", Center for Transportation Studies Rep 76-2, MIT, Cambridge, MA.
  11. Zahavi, Y. (1979). "The UMOT Project", Final Report, DOT-RSPA-DPB-20-79-3, Dept. of Transportation, Wash., DC.
  12. Deneubourg, J.L. and A. de Palma (1979). "Dynamic Models of Competition Between Transportation Modes", Envir. and Plann., Vol. 11, pp. 665–673.

Author Response

Question 1:  With respect to the earlier versions, the quality of presentation has increased. (1) Nevertheless, the scientific soundness has not increased. For instance, at the nucleus of the proposed work, there exists an extensive body of literature (a short excerpt follows at the end if it could assist the authors) that has firmly quantified on scientific grounds, even supporting a Nobel Prize on certain grounds, both the systems dynamics/ nonlinear specification of public transportation demand & supply, as well as related causal relations and cost/evaluation methods. (2) Further, anchoring public transport error on the characteristics of dominant variables, such as population must be avoided.

Response 1-1-(1):

Thank you so much for the comment. Thank you for providing the valuable references, we will study these references carefully. Although the number of the words of the revised manuscript has increased from the original 6500 to present 10300, the number of reference papers has increased from 34 to 42, and the number of figures increased from 10 to 17 and the number of tables increased from 5 to 7, there are still further and deeper work needed to be done in the future. Therefore, the valuable comment is added to the research outlook at the end of the manuscript as follows: 

Actually, the SD model has a delay phenomenon. Delay is a process whose output lags behind its input in some mode. For simplicity, this article ignores these delays and needs to be considered further in the future research. Additionally, both the systems dynamics/ nonlinear specification of public transportation demand & supply, as well as related causal relations and cost/evaluation methods need further and deeper research.

We acknowledge the fact that no single study is perfect. This study refers to many relevant literatures on system dynamics research (reference numbers are 22, 23, 24, 25, 26, 27, 28, 29, 30, 31, 32, 33, 34, 35, 36, 38 39,40), this article and these large numbers of published references have similar model structures, model simulations, model validations and similar scientific soundness, but the research question is different. The quality of work is okay, the other two reviewers also think so.

Please feel free to contact me if you have any questions. Thanks.

Response 1-1-(2):

   Thank you so much for the comment. According to the relevant literatures on system dynamics research (reference numbers are 22, 23, 24, 25, 26, 27, 28, 29, 30, 31, 32, 33, 34, 35, 36, 38 39, 40), historical data such as population, vehicle ownership are commonly used to check the model validation. Based on the model checking process of other references, we think that it is feasible to select historical population data as one of the testing data for the model test in this paper. In fact, in the revised manuscript we not only use the historical data of resident population and GDP to do the model validation, but also use the historical data of number of buses, mileage of bus network and vehicle ownership.

   In addition, one of the evaluation indicators for the construction of transit metropolis is the number of buses per capita. Residential population amount has to be considered. Therefore, it is necessary and reasonable to consider the population and other similar characteristic variables in the study of analyzing the construction effects of transit metropolis. Thanks.

Please feel free to contact me if you have any questions. Thanks.
